



# Evaluation of the LOTOS-EUROS NO₂ simulations using ground-based measurements and S5P/TROPOMI observations over Greece

Ioanna Skoulidou[1], Maria-Elissavet Koukouli[1], Astrid Manders[2], Arjo Segers[2], Dimitris Karagkiozidis[1], Myrto Gratsea[3], Dimitris Balis[1], Alkiviadis Bais[1], Evangelos Gerasopoulos[3], Trisevgeni Stavrakou[4], Jos van Geffen[5], Henk Eskes[5] and Andreas Richter[6]

[1]Laboratory of Atmospheric Physics, Aristotle University of Thessaloniki, Greece.
[2]TNO, Climate, Air and Sustainability, Utrecht, The Netherlands.
[3]Institute for Environmental Research and Sustainable Development, National Observatory of Athens, Greece
[4]Royal Belgian Institute for Space Aeronomy, Brussels, Belgium.
[5]Royal Netherlands Meteorological Institute (KNMI), De Bilt, The Netherlands.
[6]Institute of Environmental Physics and Remote Sensing, University of Bremen, Germany

*Correspondence to*: Ioanna Skoulidou (ioannans@auth.gr)

**Abstract**. The evaluation of chemical transport models, CTMs, is essential for the assessment of their performance regarding the physical and chemical parameterizations used. While regional CTMs have been widely used and evaluated over Europe, their validation over Greece is limited. In this study, we investigate the performance of the LOTOS-EUROS v2.2.001 regional chemical transport model in simulating nitrogen dioxide, NO₂, over Greece from June to December 2018. In-situ NO₂ measurements obtained from the National Air Pollution Monitoring Network are compared with surface simulations over the two major cities of Greece, Athens and Thessaloniki. The model reproduces well the spatial variability of the measured NO₂ with a spatial correlation coefficient of 0.85 for the period between June and December 2018. About half of the 14 air quality monitoring stations show a good temporal correlation to the simulations, higher than 0.6, during daytime (12-15 p.m. local time), while the corresponding biases are negative. Most stations show stronger negative biases during winter than in summer. Furthermore, the simulated tropospheric NO₂ columns are evaluated against ground-based MAX-DOAS NO₂ measurements and space-borne Sentinel 5-Precursor TROPOMI tropospheric NO₂ observations in July and December 2018. LOTOS-EUROS captures better the NO₂ temporal variability in December (0.61 and 0.81) than in July (0.50 and 0.21) when compared to the corresponding measurements of the MAX-DOAS instruments in Thessaloniki and the rural azimuth viewing direction in Athens respectively. The urban azimuth viewing direction in Athens region however shows a better correlation in July than in December (0.41 and 0.19, respectively). LOTOS-EUROS NO₂ columns over Athens and Thessaloniki agree well with the TROPOMI observations showing higher spatial correlation in July (0.95 and 0.82, respectively) than in December (0.82 and 0.66, respectively) while the relative temporal correlations are higher during winter. Overall, the comparison of the simulations with the TROPOMI observations shows a model underestimation in summer and an overestimation in winter both in Athens and Thessaloniki. Updated emissions for the simulations and model improvements when extreme values of boundary layer height are encountered are further suggested.

**Keywords**: Air quality; nitrogen dioxide; LOTOS-EUROS; Chemical transport model; Sentinel-5P; TROPOMI; NOx; emissions; MAX-DOAS; Thessaloniki; Athens; Greece

## 1 Introduction

Nitrogen oxides (NO$_x$=NO+NO₂) adversely affect human health, the environment and the ecosystems. Exposure to NO₂ is linked with high mortality rates and premature deaths (Crouse et al., 2015). NO₂ dominates the formation of ozone and inorganic aerosols in the troposphere (Seinfeld and Pandis, 1998) with detrimental effects on the climate and human health. The deposition of nitrogen leads to eutrophication and acidification (Bouwman et al., 2002). While NOx sources can be either



natural (soils, wildfires  and lightning) or anthropogenic (fossil fuel combustion, industrial emissions and emissions from road and non-road transport) (Miyazaki et al., 2017), it is estimated that human activities are responsible for 65% of the global annual NOx flux (Müller and Stavrakou, 2005).

Chemical transport models, CTMs, play an important role in air pollution assessment by providing interpretation and forecasting on air quality, based on emission inventories and atmospheric processes. CTMs are widely used serving distinct

purposes, for instance the study of regional air quality or transboundary pollution (Streets et al., 2007; Terrenoire et al., 2015) and the estimation of updated emissions (Martin et al., 2003; Müller and Stavrakou, 2005). The evaluation of CTMs is critical in order to assess the quality of the model predictions. This is achieved by comparing the model with different measurement datasets, including ground-based in-situ measurements of pollutants (Verstraeten et al., 2018) and space-borne observations (Huijnen et al., 2010).

To date, most European air quality modelling studies have focused on western and central European countries, while very few research efforts addressed Eastern Europe. Greece is located in South-eastern Europe, at the tip of the Balkan Peninsula, bordering the East Mediterranean Sea. Athens is the largest city in Greece and belongs to the administrative division of Attica that has around 3.8 million of inhabitants (EL.STAT, 2012). Air pollution in Athens is caused by the combination of high anthropogenic emissions in this densely populated area (Pateraki et al., 2013) and the particular meteorological conditions

characterized by strong  winter temperature inversions and sea breeze circulation in the summer (Kallos et al. 1993). Furthermore, the city is surrounded by mountains in the north (Parnitha, Penteli), east (Hymettos) and west (Egaleo), and the Saronicos Gulf in the south limits the dispersion of air masses above the basin (Grivas et al. 2008), see Figure 1, left.  Major NO$_x$ emission sources in the area are the large number of vehicles in circulation, the industrial area of Thriassion Plain to the west of the basin and the Mesogia Plain to the east (Fameli and Assimakopoulos, 2016), shipping and aviation. The largest

airport in Greece, Athens Eleftherios Venizelos International Airport (ATH), is also situated at the east of the basin, and also forms a local source of NOx emissions (Koulidis et al. 2020). Furthermore, the city of Piraeus in the south hosts one of the largest freight and passenger ports in Europe (Fameli & Assimakopoulos, 2016.)

Thessaloniki is the second largest city of Greece with more than 1 million inhabitants in the metropolitan area (EL.STAT, 2012) and is situated at the northern part of the country. The topography of Thessaloniki comprises of coastal (the gulf or

Saronikos) and mountainous areas (the mountain of Hortiatis), see Figure 1, right. Similarly to Athens, particular meteorological features such as sea and land breeze and valley mountain winds, affect the air quality of the city (Moussiopoulos et al., 2009). According to Poupkou et al. (2011), road transport and industrial emissions are the two main sources of NOx emissions at the greater area of Thessaloniki while the majority of industrial activity is concentrated at the west and north-west part of the city. Furthermore, at the east of the city, the second largest airport of Greece, Macedonia International Airport

(SKG), is situated. Notably, transboundary pollution has also been observed from lignite-burning power plants  in central and eastern Europe (Zerefos et al., 2000.)

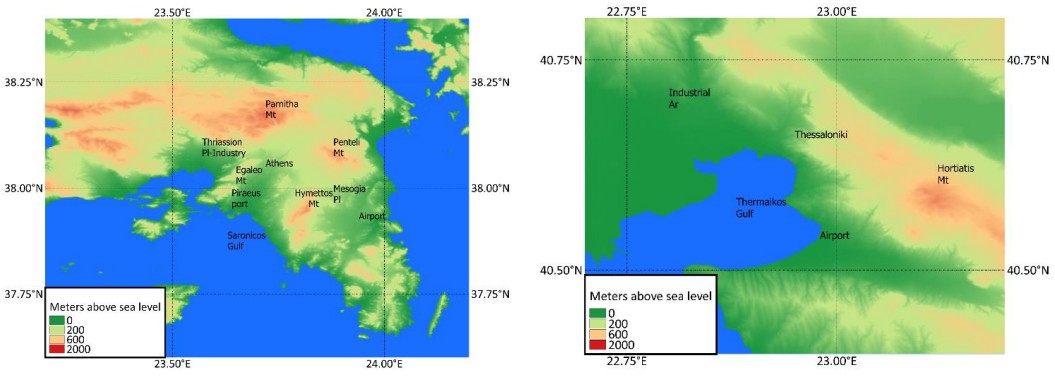



**Figure 1: The topographic features that affect air pollution as well as the areas of known NOx sources in Athens (left) and Thessaloniki (right); Mt refers to Mountain, Pl to Plain and Ar to Area [Created using https://land.copernicus.eu/imagery-in-situ/eu-dem/eu-dem-v1.1 and QGIS software].**

In this work, the $NO_2$ simulations of the LOTOS-EUROS CTM are evaluated over the region of Greece through detailed comparisons with ground-based measurements and space-borne observations. To our knowledge, the performance of the LOTOS-EUROS model has not been evaluated over Greece before. In-situ $NO_2$ surface measurements over the regions of Athens and Thessaloniki are obtained for the time period between June and December 2018 and are compared against the surface simulations of the model. The availability of the measurements before June is sparse and so the study is conducted for the period between June and December. Additionally, $NO_2$ tropospheric columns simulated by the model are compared with $NO_2$ columns retrieved from MAX-DOAS systems and satellite retrievals from Sentinel5 Precursor/TROPOMI in Athens and Thessaloniki. This study is structured as follows; in section 2.1 the model and the adopted setup are described; in 2.2 and 2.3 the datasets used for the evaluation are presented. In section 3 we discuss the comparisons of the simulations with the aforementioned datasets. In section 3.1 the simulations are compared against in-situ measurements, section 3.2 presents the results of the MAX-DOAS and LOTOS-EUROS column comparisons, and in section 3.3 the S5P/TROPOMI retrievals are used to further evaluate the $NO_2$ simulated columns. Finally, conclusions are presented in section 4.

## 2 Data and methodology

The performance of the LOTOS-EUROS model is studied over the region of Greece, and its surrounding neighbouring countries, as well as in more detail around the two largest cities in Greece, namely Athens [37.9838° N, 23.7275° E] and Thessaloniki [40.7369° N, 22.9202° E] (see Figure S1). For this study Athens refers to the Attica basin which includes the city of Piraeus and the suburbs, while Thessaloniki refers to the whole metropolitan area of Thessaloniki.

### 2.1 The LOTOS-EUROS CTM

In this study we used the open source CTM LOTOS-EUROS v2.2.001 (Manders et al. 2017, https://lotos-euros.tno.nl/). The model can simulate distinct components (i.e. oxidants, secondary inorganic aerosols, primary aerosol and heavy metals) in the troposphere (Schaap et al., 2008). It has been extensively used in the past for air quality studies and forecasting. In particular, a good agreement was reported between $NO_2$ LOTOS-EUROS simulations and MAX-DOAS observations in the Netherlands (Vlemmix et al., 2015). Schaap et al. (2013) used the $NO_2$ model results to study the sensitivity of the retrieved $NO_2$ columns from the OMI/Aura satellite instrument to anthropogenic emissions pointing out the need of model simulations along with satellite observations in order to assess emission trends. Curier et al. (2014) used $NO_2$ retrieved from OMI and LOTOS-EUROS to determine the $NO_2$ trends from 2005 to 2010 over Europe and found significant decreases in industrialized areas. LOTOS-EUROS constitutes one of the state-of-the-art atmospheric chemistry models used by the Copernicus Atmosphere Monitoring Service (CAMS, www.copernicus-atmosphere.eu) to provide daily forecasts of the main air pollutants (i.e. ozone, $NO_2$ and $PM_{10}$). These simulations have recently been used in an effort to quantify the effects of the lockdown due to the COVID-19 pandemic over Greece as observed by S5P/TROPOMI (Koukouli et al., 2020).

In this study we performed two model simulations. The first one, the outer area, includes Central and Southern Europe (15° W to 45° E and 30° – 60° N) with a horizontal resolution of 0.25°x0.25° for the year 2018 (referred to hereafter as the *European domain*), see Figure S1. The second run consists of an inner area of the coarser European domain and covers the period between June and December of 2018. The smaller domain in this case spans from 33° to 46° North and 18° to 29° East with a grid resolution following the resolution of the emission inventory used for our setup (0.1° longitude x 0.05° latitude), referred to hereafter as the *Greek domain*. Both simulations were driven by the operational meteorological data from the European Centre for Medium-Range Weather Forecasts (ECMWF) with a horizontal resolution of 7 km × 7 km and a temporal resolution of 1 hour for the surface variables and 3 hours for the multi-level parameters (Flemming et al., 2009). The 10 vertical levels used





for the simulations of the atmospheric components were defined as a coarsening of the meteorological model levels and are spanning the troposphere from the surface to a top around 175 hPa (about 12 km). The anthropogenic emission inventory used is the CAMS-REG (CAMS Regional European emissions) version 2 for the year 2015 at 0.1°×0.05° (Granier et al., 2019; Kuenen et al., 2014). Biogenic emissions (isoprene) are calculated online using the meteorology and a detailed land use and tree-species database. Soil NO emissions are taken from a parametrization depending on soil type and soil temperature (Novak and Pierce, 1993) while NOx production from lightening is not included in the model. The aggregated total NO emissions from anthropogenic and biogenic sources used from June to December 2018 are shown in Figure S2 next to the biogenic NO emissions for the same period. Biogenic NO emissions constitute 11% of the total NO emissions in the area as seen in Figure S2, which shows Greece, south Albania, south North Macedonia, south Bulgaria and west Turkey, while the 97% of the NOx emissions is emitted as NO and the rest as $NO_2$ in the model.

The initial and boundary conditions for the European domain are obtained from the Copernicus Atmosphere Monitoring Service (CAMS, https://atmosphere.copernicus.eu/). The CAMS global near-real time (NRT) product is used for the gas and aerosol concentrations with a spatial resolution of 35 km × 35 km and a temporal resolution of 3 hours. The initial and boundary conditions used in the case of the Greek domain are provided from the outputs of the coarser European domain, while the top boundary conditions used in both domains are obtained from the CAMS-NRT as well. The gas phase chemistry of the model is described using a modified version of Carbon Bond Mechanism IV (CBM-IV) scheme (Gery et al., 1989), while the aerosol chemistry is represented by the ISORROPIA II (Fountoukis and Nenes, 2007). More details on the chemistry module of LOTOS-EUROS can be found in Manders et al. (2017). For the biomass burning emissions and wildfires, the Global Fire Assimilation System (GFAS), that assimilates fire radiative power (FRP) observations from satellite-based sensors (Kaiser et al., 2012), is used in the LOTOS-EUROS simulations. In the Greek domain and during the period of study some artificial fires were detected but since no big wildfires have been recorded in the area, these GFAS inventory was not taken into account in the nested simulation.

**2.2 Ground-based measurements**

In order to validate the $NO_2$ simulations derived from LOTOS-EUROS over Greece we compare model-derived surface concentrations with in-situ air quality measurements performed in the regions of Athens and Thessaloniki. Furthermore, the simulated $NO_2$ tropospheric columns over the Greek domains are compared against MAX-DOAS $NO_2$ columns, also situated in Athens and Thessaloniki.

**2.2.1 In-situ $NO_2$ measurements**

Hourly in-situ measurements of $NO_2$ concentrations over the Greek domain were obtained from the National Air Pollution Monitoring Network (http://www.ypeka.gr/) from June to December 2018. The aforementioned data are routinely reported to the European Environmental Agency Air Quality database. A chemiluminescence method is used for the measurement of nitrogen dioxide concentrations at the stations. The stations in Athens are operated by the Department of Air Quality while the rest of the Greek stations are operated by regional administrations. Hourly $NO_2$ measurements are available for Athens and Thessaloniki. The stations used for the evaluation of the model over Greece, were selected carefully in order to be well distributed and to be representative of the local emission sources. The locations of the stations are given in Figure 2 for Thessaloniki (top) and Athens (bottom). The marked colours over the stations refer to the average $NO_2$ measured at each station between June and December 2018. We should acknowledge possible representativity errors when comparing the measurements from urban traffic stations with the mean value of a model grid cell (0.1°x0.05°) (Blond et al., 2007). For this reason, stations characterized as urban traffic stations, localised close to busy traffic roads of the city and showing very large values, are excluded from the validation. As a result, out of a total of 24 stations reporting to the repository, 5 and 9 stations for the region of Thessaloniki and Athens, respectively are retained for the model evaluation. Measurements from these stations





were used in the past to investigate the NOx trends in Athens (Mavroidis and Ilia, 2012) to assess the impact of the economic crisis in Greece after 2008 (Vrekoussis et al. 2013), and to reveal how the surface $NO_2$ concentrations are reflected on the OMI/Aura retrieval (Zyrichidou et al., 2013).

### 2.2.2 MAX-DOAS measurements

In this study, tropospheric $NO_2$ columnar measurements from Multi Axis Differential Optical Absorption Spectroscopy (MAX-DOAS) systems located in Thessaloniki and Athens are compared with LOTOS-EUROS simulated columns for July and December 2018. For the region of Thessaloniki, *Phaethon*, a miniature spectrometer ground-based MAX-DOAS system, is used. The system was developed in 2006 at the Laboratory of Atmospheric Physics (LAP) in Thessaloniki, Greece (Kouremeti et al., 2008). This system operates regularly on the roof of the Physics Department of the Aristotle University campus which is located in the centre of Thessaloniki (Drosoglou et al., 2017). At the same location, an air quality measuring station, labelled *AUTH* in Figure 2, is in operation by the Region on Central Macedonia. For this study, we used MAX-DOAS observations at 15° elevation angle in order to avoid uncertainties introduced due to aerosols at lower elevation angles (Sinreich et al., 2005) and at two azimuth angles: 220° and 255° designated in Figure 2 by the purple lines 1 and 2, respectively. In these viewing directions the MAX-DOAS system probes air over the centre of the city and the gulf of Thessaloniki, an area which is usually supplied by air from the western part and the industrial area of the city, i.e. directly from the urban environment. The average tropospheric columns from the two azimuth angles was calculated since both directions fall into the same grid pixel of the model simulations. The retrievals are based on geometrically approximated Air Mass Factor (AMF) (Wagner et al., 2010). The tropospheric $NO_2$ derived from MAX-DOAS instruments positioned at three different locations around Thessaloniki and the OMI/Aura satellite were compared during a 6-month campaign showing good agreement over the rural and the suburban areas (Drosoglou et al., 2017).

In Figure 2 (bottom) the location of the MAX-DOAS instrument used for the comparisons of the $NO_2$ tropospheric column in Athens is marked. The MAX-DOAS instrument is installed at northeast of Athens, at Penteli mountain (527 m above sea level), and belongs to the BREDOM network (Bremian DOAS network for atmospheric measurements), https://www.iup.uni-bremen.de/doas/ (Gratsea et al., 2016). Two azimuthal viewing angles are selected in this case as well, at 120° and at 232.5°, and are represented by the purple lines in Figure 2. The first one, marked with "R" is characterized as a rural unobstructed direction, while the other one is named "U" and views towards an urban direction (Figure 2). The azimuthal viewing angles selected are representative for urban and rural air quality environment conditions in Athens. The tropospheric $NO_2$ vertical columns were derived, as in the case of Thessaloniki, using the geometric approximation.

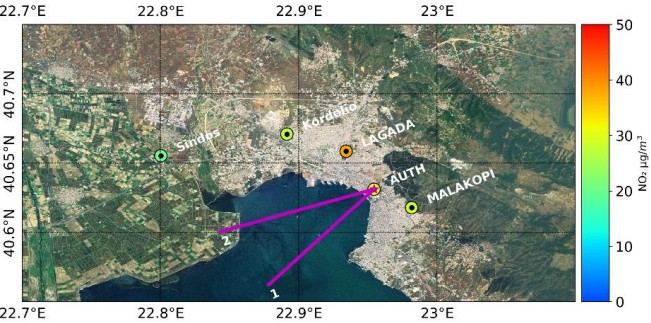





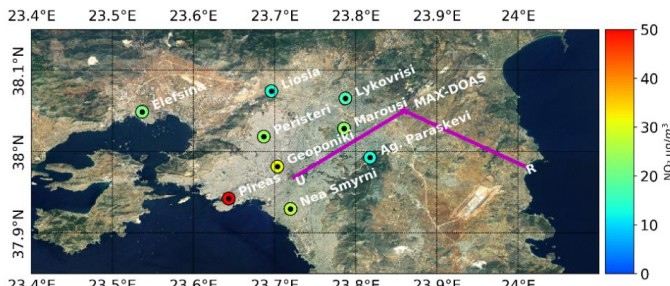

**Figure 2: In-situ air quality measurements in the region of Thessaloniki (top) and the region of Attica (bottom). The points represent the stations while the colour bar denotes the average NO₂ measurements over June- December 2018. The purple lines represent the direction of the MAX-DOAS measurements in the area [Created using background from ArcGIS service].**

**2.3 S5P/TROPOMI NO₂ observations**

The Sentinel-5 Precursor, S5P, satellite was launched on October 13$^{th}$, 2017, carrying the TROPOspheric Monitoring Instrument, TROPOMI (Veefkind et al., 2012). The satellite flies in a near-polar, sun-synchronous orbit in an altitude of 824 km with an equatorial crossing at 13:30 local solar time (LST). TROPOMI is a passive, nadir-viewing spectrometer measuring wavelengths between the ultraviolet and the shortwave infrared. The swath width of TROPOMI in the Earth's

surface is approximately 2600 km and the ground pixel of the instrument at nadir is $7{\times}3.5km^2$, $5.5{\times}3.5km^2$ since August 2019, achieving near global coverage in one day. The wavelength range used for the NO₂ column retrieval algorithm is between 405 and 465 nm, while detailed information on the algorithm and the data can be found in the TROPOMI NO₂ Algorithm Theoretical Basis Document (van Geffen et al., 2019, 2020). The data are constantly validated by the Mission Performance Center Validation Data Analysis Facility[1], VDAF, as well as several TROPOMI NO₂ validation papers that have been recently

submitted (Judd et al., 2020; Verhoelst et al., 2020). Compared with globally deployed ground-based remote sensing MAX-DOAS instruments, the TROPOMI tropospheric NO₂ shows on average 30% lower levels (S5P MPC Routine Operations Consolidated Validation Report, 2020). The validation work also shows high correlations with the independent observations and a nearly linear scaling of the error with the tropospheric column amount. Part of the bias is attributed to retrieval inputs, in particular cloud pressure retrievals and the albedo used. But a second part is attributed to the a-priori, which is available

globally at a resolution of 1x1 degree. This is not enough to resolve the NO₂ profiles near point sources and over cities. As a result the tropospheric column is often underestimated at the hotspots (and somewhat underestimated in rural regions). Note, however that because the averaging kernels are used in our case, the comparison with LOTOS-EUROS is not influenced by the retrieval a-priori (Eskes and Boersma, 2003). Therefore, we expect a TROPOMI low bias of the order of 10-20% to remain, influencing the comparisons.

In this study we used the reprocessed daily data, RPRO, version 01.02.02 of TROPOMI for July and the offline data, OFFL, for December 2018, which can be obtained via the Copernicus Open Data Access Hub (https://s5phub.copernicus.eu/). The data are filtered with a quality assurance value qa_value>0.75, ensuring mostly cloud-free observations, and gridded onto the LOTOS-EUROS grid at 0.1°×0.05°. The TROPOMI tropospheric vertical column is compared with the simulated NO₂ column derived from the LOTOS-EUROS after the averaging kernel of the TROPOMI vertical columns is applied to the simulations

in order to ensure a consistent comparison between the modelled and measured columns (Eskes and Boersma, 2003).

---

[1] http://mpc-vdaf.tropomi.eu/



## 3 Results

### 3.1 LOTOS-EUROS and in-situ measurements

In the following, we compare the hourly modeled surface $NO_2$ concentrations from the lowest layer of the simulations with hourly in-situ measurements. Table 1 and Table 2 summarize the direct comparisons of LOTOS-EUROS $NO_2$ surface
concentrations to $NO_2$ concentrations measured at 5 stations in Thessaloniki and 9 in Athens, respectively. The stations are characterized by their type, here we distinguish 5 different types: traffic, urban background, urban industrial, suburban background and suburban industrial. The correlation coefficients based on the hourly values are calculated for five different time periods; the whole period from *June to December* 2018, the *winter* period where measurements from November and December are only taken into account, the *summer* period that includes July and August, the *day* period that includes only the
daytime data between 12 p.m. and 15 p.m. local time during the whole period, and finally the *night* period that refers only to the hours between 0 a.m. and 3 a.m. local time during the whole period. The distinct periods are selected in order to study the seasonal (summer and winter) and diurnal (day and night) performance of the model when different parameters may affect the simulations. For instance the atmospheric mixing and the parameterization of boundary layer pollutants are mostly affected by the diurnal cycle (Nester and Fiedler, 1992).

In the region of Thessaloniki (Table 1) the correlation coefficients calculated over June to December range from r = 0.49 to r = 0.58. Overall, the mean correlation in summer is slightly lower (0.49) than in winter (0.55). During daytime the correlation generally increases by 13-21% and decreases by 27-34% during night in comparison to the whole period except for the urban industrial station "Sindos" where it decreases by about 20% during daytime and it slightly increases at night. In the case of the two urban background stations, "Malakopi" and "AUTH" the correlations are very good (r=0.69 and 0.63, respectively) during
daytime.

In the region of Athens (Table 2) the same calculations are performed for the 9 selected stations. The average correlation over June to December is ~0.5 with two suburban background stations ("Ag. Paraskevi" and "Liosia") having the lowest values (r=0.39 and r=0.34 respectively) and the urban background/traffic stations ("N. Smurni" and "Marousi") the highest ones (r=0.62). In this case, a clear seasonal pattern in the model's performance, as is the case for Thessaloniki, was not found. On
the other hand, when limiting the analysis to the day time period, the majority of the correlations are improved compared to the June-December period and increased by about 5-55% and 70% for the suburban background station "Liosia". The urban background station "Nea Smurni" and the urban traffic station near the port of Piraeus exhibit decreased correlations during day in comparison with the June to December period (-29% and -55% respectively). Finally, the correlations worsen during nighttime (decreased by about 9-56% compared to the whole period) in Athens except for the "Piraeus" urban traffic station,
which is located in the city port (Table 2. The correlation at the suburban industrial station "Geoponiki" during daytime reaches 0.72, compared to 0.44 for the nightime period.

**Table 1. Correlations and RMSE values (in parenthesis) between the NO₂ surface observations at the in-situ stations in Thessaloniki and the surface concentrations calculated at the corresponding model grid cell. Correlations higher than 0.6 are marked in bold while the correlations lower than 0.4 are marked in italic.**

| Station name | Type of station | Correlation coefficient and RMSE in parenthesis | | | | | | | | |
|---|---|---|---|---|---|---|---|---|---|---|
| | | J-D | | Summer | | Winter | | Day | | Night | |
| Lagada | Traffic | 0.49 | (25.36) | 0.46 | (15.74) | 0.45 | (33.19) | 0.58 | (24.80) | *0.34* | (27.97) |
| Malakopi | Urban background | 0.58 | (18.32) | 0.57 | (17.62) | **0.63** | (17.33) | **0.69** | (14.80) | 0.40 | (18.78) |
| AUTH | Urban background | 0.52 | (21.93) | 0.47 | (17.95) | **0.62** | (23.70) | **0.63** | (19.66) | *0.34* | (19.93) |





| Kordelio | Urban industrial | 0.52 | (18.89) | 0.46 | (15.78) | 0.51 | (21.70) | 0.59 | (13.10) | *0.38* | (17.71) |
| Sindos | Urban industrial | 0.54 | (14.85) | 0.49 | (14.17) | 0.53 | (14.66) | 0.43 | (8.01) | 0.57 | (15.72) |


**Table 2. Correlations and RMSE values (in parenthesis) between the NO₂ surface observations at the in-situ stations in Athens area and the surface concentrations calculated at the corresponding grid pixel of the simulations. The correlations higher than 0.6 are marked in bold while the correlations lower than 0.4 are marked in italic.**

| Station name | Type of station | Correlation coefficient and RMSE in parenthesis | | | | | | | | | |
| | | J-D | | Summer | | Winter | | Day | | Night | |
| Geoponiki | Suburban industrial | 0.60 | (21.99) | 0.58 | (23.95) | 0.60 | (19.33) | **0.72** | (13.16) | 0.44 | (26.43) |
| Liosia | Suburban background | *0.34* | (16.42) | *0.35* | (18.25) | *0.33* | (14.85) | 0.59 | (7.58) | *0.15* | (17.58) |
| Lykovrisi | Suburban background | 0.59 | (15.62) | 0.52 | (17.32) | **0.66** | (12.17) | **0.62** | (8.63) | 0.52 | (15.26) |
| Marousi | Urban traffic | **0.62** | (20.24) | **0.61** | (22.70) | **0.62** | (16.80) | **0.67** | (8.78) | 0.48 | (24.92) |
| Nea Smurni | Urban background | **0.62** | (20.43) | **0.67** | (20.63) | 0.50 | (21.10) | 0.44 | (12.47) | 0.53 | (24.74) |
| Pireas | Urban traffic | 0.41 | (37.82) | 0.45 | (42.05) | 0.42 | (29.89) | *0.18* | (50.05) | 0.59 | (23.82) |
| Peristeri | Urban background | 0.55 | (20.87) | **0.61** | (23.97) | 0.54 | (19.24) | **0.61** | 12.05 | 0.42 | (25.42) |
| Elefsina | Suburban industrial | 0.51 | (19.77) | 0.42 | (25.54) | 0.56 | (14.88) | 0.50 | 10.93 | 0.46 | (28.96) |
| Ag. Paraskevi | Suburban background | *0.32* | (19.13) | *0.27* | (19.24) | *0.32* | (19.45) | 0.50 | 6.90 | *0.18* | (23.62) |


In Figure 3 the mean NO₂ simulations for each period chosen are compared against the NO₂ measurements for all 14 stations and in Table 3 the respective statistics are given. Each period is marked with a different color: winter in blue, summer in orange, daytime in red, nighttime in purple and June-to-December in green. Overall, and irrespective of the temporal choice, the simulations are found to underestimate the in-situ measurements, as shown by the linear regression slopes, colored in tandem to the datasets. The model behaves similarly in winter (0.36 and 14.61 μg.m⁻³ the slope and the offset of the regression line respectively) and summer (0.43 and 10.96 μg.m⁻³ the slope and the offset respectively), while the spatial variability is better reproduced in summer when the spatial correlation coefficient is 0.86. The difference between day and night comparisons, shown in Table 1 and Table 2 is evident whereas the strong model underestimation in daytime (0.14 and 5.55 μg.m⁻³ the slope and the offset respectively) merits further analysis. During the night period the model overestimates the measurements of low NO₂ and underestimates the higher concentrations (0.63 and 9.85 μg.m⁻³ the slope and the offset respectively). The spatial correlation is higher during day (0.80) than in night (0.70).


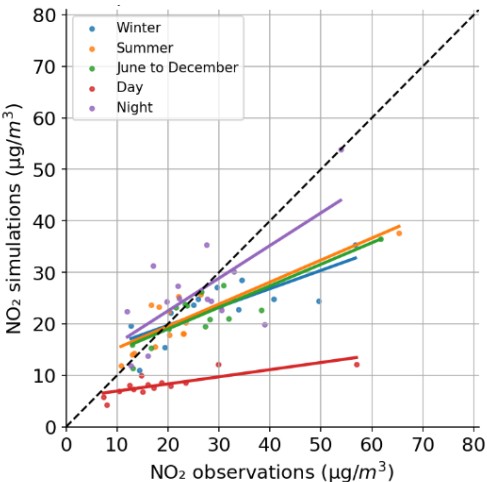

**Figure 3 Scatterplot of the simulated and observed NO₂ concentrations for the 14 stations used for the validation at the different periods. Winter is the November and December months (blue line), summer July and August (orange line), June to December includes measurements for the whole period of study (green line), day represents the daily hours (between 12 to 15 p.m. local time) during the whole period (red line) and night the night hours (between 0 and 3 a.m. local time) during the whole period (purple line).**


**Table 3 The coefficients of the scatterplot in Figure 3 between the averaged NO₂ simulations and the in-situ measurements for the 14 stations used for the validation at the different periods; June to December, winter, summer, daytime and night-time.**

| Period | Correlation coefficient | Slope | Offset ($\mu g.m^{-3}$) |
|---|---|---|---|
| Winter | 0.78 | 0.36 | 12.61 |
| Summer | 0.86 | 0.43 | 10.96 |
| June to December | 0.85 | 0.42 | 10.62 |
| Day | 0.80 | 0.14 | 5.55 |
| Night | 0.70 | 0.63 | 9.85 |

To investigate the performance of LOTOS-EUROS simulations in greater detail we compared the hourly NO₂ concentrations

at the urban background representative stations "Malakopi" in Thessaloniki and "Peristeri" in Athens. Time series for these

stations are shown in Figure 4. For both stations the LOTOS-EUROS simulations follow the measurements satisfactorily.

During some periods the NO₂ simulations are low compared to the measurements and this in most cases coincides with the

daytime underestimation of the model, as shown at Figure 3 as well. This occurs, for instance, at the "Malakopi" station (upper

panel) during the period of 1 June to 15 July and at the "Peristeri" station in early June. On the other hand, there are some days

for which much higher NO₂ levels are simulated than observed, mainly during nighttime, as seen for example during July at

the "Peristeri" station (lower panel).

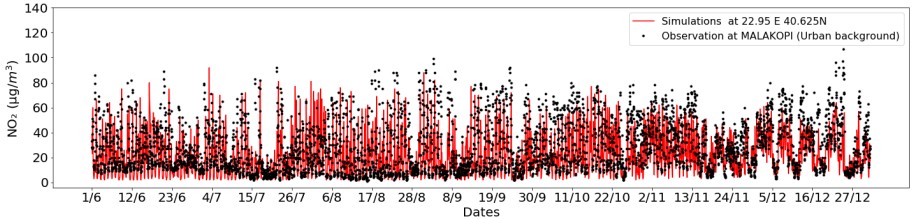





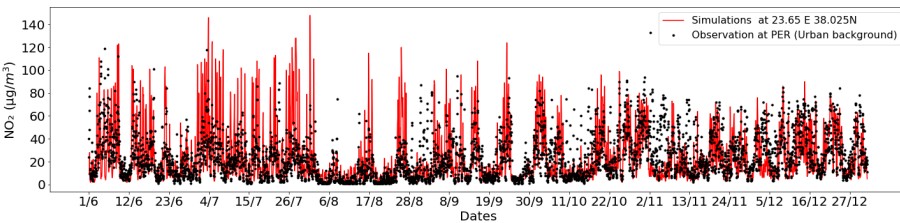

**Figure 4: Time series of the hourly in-situ measurements at two stations (black points) and the simulations of the model in the corresponding pixel (red lines) during June 2018 and December 2018. The urban background "Malakopi" station in Thessaloniki (top) and the urban background "Peristeri" station in Athens (bottom) are shown in the figure.**

The relative biases between the simulated and measured values of the each periods are shown in the box and whisker plot in Figure S3. The period over June to December (green box) shows a range of biases between -40% and 23% showing an underestimation of the measurements in most cases and a median bias of -10%. The relative biases in winter (blue box) range between -50% and 8% showing a clear underestimation of the measurements, while Ag. Paraskevi has a high positive bias of 54%. In summer though, the median relative bias is -2% and the model both underestimates and overestimates the measurements almost equally. The strong underestimations of the measured $NO_2$ during day hours is depicted by the high negative biases at the daytime period and the absence of positive biases (red box). The station "Piraeus" shows a very low bias (-78%) during daytime which can be explained by the very high pollutant levels emitted near the station due to traffic and shipping. The night simulations show very high overestimation at the suburban industrial and background stations of "Elefsina" and "Ag Paraskevi" respectively.

From this comparison we found that LOTOS-EUROS $NO_2$ surface simulations are biased compared to the in-situ measurements over the two major cities of Greece over June to December 2018 showing an underestimation of the measurements with a mean relative bias of -13%, a median relative bias of -10%, a high spatial correlation coefficient equal to 0.85 and an average temporal correlation of 0.52. The separate evaluation during distinct periods of time shows that the model underestimates the $NO_2$ surface concentrations mostly during daytime (12 to 15 pm local time) and overestimates the low concentrations during the night-time (0 to 3 am local time). The daytime underprediction could be partly due to representation issues related to the location of the stations, which lie near urban city centres and industrial areas, that cannot be well resolved by the model at 0.10°x0.05° (Liu et al., 2018) as in the case of "Piraeus" station near the port in Athens. Further, the daytime boundary layer height provided is likely too high, thus resulting in low $NO_2$ surface concentrations (Huijnen et al. 2010). Moreover, the chemistry of a model is indirectly affected by the photolysis rate and the meteorology (as an example the solar radiation is affected by the cloud coverage which in our case hourly cloud coverage data are obtained from ECMWF). However, the mismatch between the simulations and the measurements is found to be more significant during night-time when the model strongly overestimates in some cases the surface observations, the spatial correlation coefficient is lower (0.80 during day and 0.70 during night) and the temporal correlation coefficient is much lower than in daytime as well. This could be due to possible flaws in the representation of the boundary layer and can be explained by a very low boundary layer height adoption during the night and a small vertical mixing as well (Bessagnet et al., 2016). The boundary layer height in LOTOS-EUROS is taken from the ECMWF operational weather analysis data and is based on the bulk Richardson number following the conclusions of the Seidel et al. (2012) review. Lampe (2009) further showed that during night the urban heat island can cause a larger boundary layer height and a stronger mixing that leads to the decrease of surface pollutants levels. We find that the model shows a slight dependency on the season underestimating the $NO_2$ during winter at most stations (average relative bias -15%) while in summer the average relative bias is -1% but with a larger range of the biases. The negative and positive biases can be further explained by the underestimation, or the overestimation of the anthropogenic NOx emissions


used in the model as they refer to the year 2015. Moreover the default time profiles of the emissions are used and can cause

representativeness issues for the case of Greece, adding further biases.

### 3.2 Comparison with MAX-DOAS observations

Figure 5 shows time series of the tropospheric $NO_2$ vertical column density from the MAX-DOAS system in the Aristotle University of Thessaloniki (AUTH) and the simulated $NO_2$ tropospheric column from LOTOS-EUROS for July and December. Only solar zenith angles lower than 75 degrees are considered for the comparisons and less measurements are available in

December than in July (176 and 248, respectively). The MAX-DOAS in the center of Thessaloniki observes high $NO_2$ columns during the winter months and lower levels during the spring season, similar to the observations shown in Drosoglou et al. (2017) for the year 2014-2015. In July (hours between 6 a.m. and 13 p.m. UTC), the measured and simulated columns show a good agreement while in winter (hours between 6 a.m. and 13 p.m. UTC) the measured columns are in many cases higher than the simulated ones. The mean $NO_2$ observations in July and December are $5.02\pm3.59$ and $12.54\pm7.83\times10^{15}$ molec.cm$^{-2}$ ,

respectively, while the mean model column over the same periods are $4.60\pm3.19$ and $8.43\pm4.12\times10^{15}$ molec.cm$^{-2}$ (Table 4). In both seasons the bias is negative, higher in December, about -33% ($-4.11\times10^{15}$ molec.cm$^{-2}$), and lower in July, about -8% ($-0.42\times10^{15}$ molec.cm$^{-2}$) (Table 4). The daily mean correlation in December is 22% higher than in July (0.61 and 0.50 respectively). The model shows similar characteristics for $NO_2$ columns and surface concentrations when compared with MAX-DOAS and ground observations respectively with a negative bias in AUTH for both summer (-8.44% and -20.8%

respectively) and winter (-32.75% and -39.2% respectively).

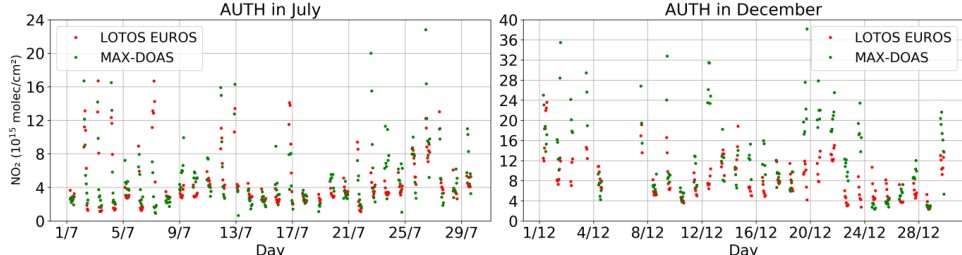

**Figure 5** Time series of LOTOS-EUROS (red) and MAX-DOAS (green) $NO_2$ columns over AUTH for July (left) and December (right) 2018.

**Table 4** Statistics between the MAX-DOAS observations and the LOTOS-EUROS simulations for July and December in Thessaloniki. Columns are expressed in $10^{15}$ molec.cm$^{-2}$.

| Month | MAX-DOAS mean | Standard deviation | LOTOS-EUROS mean | Standard deviation | Bias (relative bias) | Correlation coef. |
|---|---|---|---|---|---|---|
| July | 5.02 | 3.59 | 4.60 | 3.19 | -0.42 (-8.44%) | 0.50 |
| December | 12.54 | 7.83 | 8.43 | 4.12 | -4.11 (-32.75%) | 0.61 |


Scatter plots of the daily average $NO_2$ columns from the MAX-DOAS against the LOTOS-EUROS simulation for July (left) and December (right) are shown in Figure 6. Linear regression lines and equations are given along with the plots at the top. For both July and December the regression lines show that the model overestimates the low and underestimates the high values of $NO_2$ columns. This behavior is more noticeable during December where the slope of the line is as small as 0.33 and the

offset is equal to $4.46\times10^{15}$ molec.cm$^{-2}$. In July the slope and the offset are 0.46 and $2.30\times10^{15}$ molec.cm$^{-2}$, respectively.





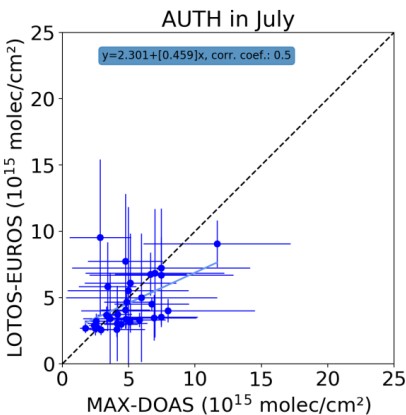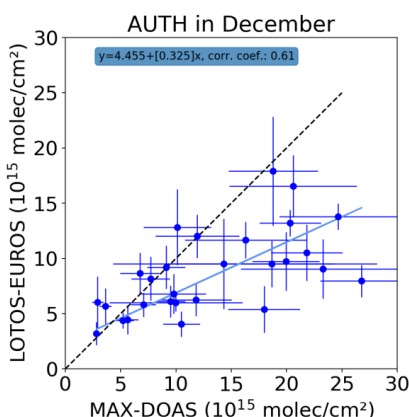

**Figure 6: Scatter plot between daily mean LOTOS-EUROS and MAX-DOAS NO₂ columns in AUTH for July (left) and December (right). The regression equation and the correlation coefficient between the model and the observed data are given in the top right of each plot.**

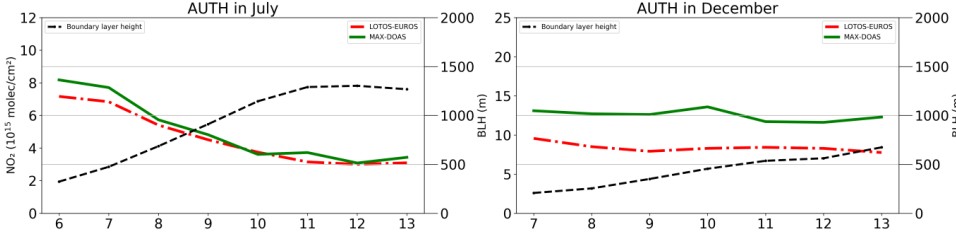

**Figure 7: Average diurnal cycle of the MAX-DOAS (green line), LOTOS-EUROS (red dashed line) NO₂ columns and boundary layer height (black dashed line) during July (left) and December (right) in Thessaloniki.**

Comparisons of the average diurnal cycles for July and December are shown in Figure 7. Overall, LOTOS-EUROS reproduces very well the diurnal cycle in July with the highest values between 6 and 7 UTC in the morning. On the other hand, the modelled NO₂ levels in December are about 32% lower than the MAX-DOAS columns. As expected, the NO₂ measurements are higher in winter than in summer because of the higher emissions in winter and the strong photochemical loss of NO₂ in summer (Boersma et al., 2009). The MAX-DOAS columns show a small peak at 10 UTC while LOTOS-EUROS shows a quite constant diurnal cycle. The lower model values in December could be partly explained by the fact that the NOx modelled lifetime may be too short, due to underestimated NOx emissions or because of a low boundary layer. Other parameters that play a pivotal role in the measurements and the simulations are related to meteorology, such as temperature and cloud coverage. According to Schaub et al. (2007), high temperatures and longer days result in shorter NOx lifetime compared to lower temperatures and less hours of daylight, while they also showed that a cloud fraction of 0.2 results in a longer NOx lifetime than a cloud fraction of 0.1, as a result of the decreased amount of solar radiation caused by higher cloud fraction. Therefore, uncertainties in the meteorological input data (cloudiness and temperature) in the model may induce uncertainties in the photochemical conversion and lifetime of NOₓ.

The same procedure followed for the case of Thessaloniki is also followed for the MAX-DOAS in Athens; the two distinct azimuthal angles have been selected; the azimuthal viewing angle towards the urban area (U) and the azimuthal viewing angle towards rural area (R). Since the MAX-DOAS instrument in Athens is located in a mountainous area around 500 m above sea level and in order to succeed consistency in the comparison between the measurements and the simulations, we integrated the modelled NO₂ columns above the model altitude of about 424 m. Figure 8 shows the time series of the tropospheric NO₂ vertical column density from the MAX-DOAS in Athens and the simulated NO₂ tropospheric columns from LOTOS-EUROS (above 424 m) at the corresponding model grid cells for July (left) and December (right) between 6 a.m. and 13 p.m. in the





urban direction, while for the rural direction is shown in Figure S4. The model underestimates slightly the measurements for both periods at the urban direction, similar to Thessaloniki, and stronger at the rural direction.

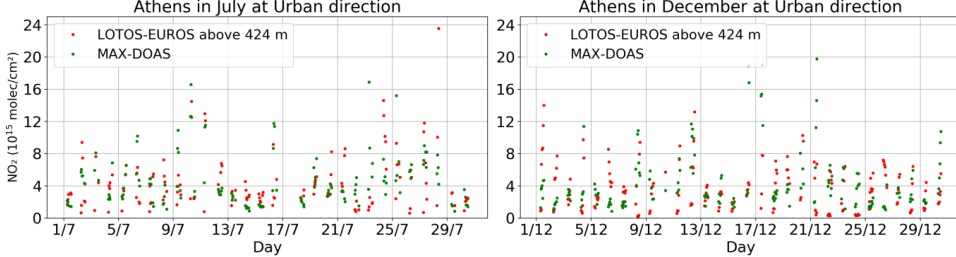

**Figure 8: Time series of LOTOS-EUROS (red) and MAX-DOAS (green) NO₂ columns over Athens for July (left) and December (right) 2018 in urban direction.**

This is further confirmed by the statistics in Table 5 for the urban and rural directions. The average NO₂ tropospheric columns measured for July are 4.44±3.11 and 1.87±1.94 ($10^{15}$ molec.cm⁻²) while the simulated columns are 4.34±3.77 and 1.19±1.30 ($10^{15}$ molec.cm⁻²) for the urban and rural directions respectively. The model slightly underestimates the NO₂ columns in July at the urban direction with a relative bias of -2.23% while at the rural direction indicates a stronger underestimation and the bias is -36.48%. In December the biases show similar characteristics as in July at both the urban and the rural directions and

underestimate the measurements (-14.48% and -26.78% respectively). Similar biases were seen for the comparison of surface simulations with ground based observations that are available in the same model cell pixel as the urban direction MAX-DOAS measurement (-2.91% for the summer period and -17.16% for the winter period). The daily mean correlation in July between the integrated columns above 424 m and the corresponding observations at the urban and rural directions are 0.41 and 0.21 respectively, while the correlations found with the full profile simulations are higher and equal to 0.56 and 0.42 for the two

directions showing that the full profile follows better the variability of the observations, as seen in Figure 9 and Figure S5. The daily mean correlation between the measurements and the partial columns above 424 m is low in comparison to the full profile of LOTOS-EUROS columns in December as well (0.19 and 0.41, respectively) at the urban direction while for the rural direction increases and is slightly higher for the integrated simulation above 424 m (0.81) compared to the full profile (0.79). The number of available observations is higher in winter than in summer in the case of Athens and for the urban direction is

155 for summer and 181 for winter and for the rural direction 90 and 194 measurements were used respectively.

**Table 5 Statistics between the MAX-DOAS observations and the LOTOS-EUROS simulations for July and December in Athens at the urban and rural azimuthal directions. Columns are expressed in $10^{15}$ molec.cm⁻².**

|  | MAX-DOAS mean | Standard deviation | LOTOS-EUROS mean | Standard deviation | Bias (relative bias) | Correlation coef. |
|---|---|---|---|---|---|---|
| July (urban) | 4.44 | 3.11 | 4.34 | 3.77 | -0.10 (-2.23%) | 0.41 |
| December (urban) | 4.12 | 3.55 | 3.53 | 2.77 | -0.60 (-14.48%) | 0.19 |
| July (rural) | 1.87 | 1.94 | 1.19 | 1.30 | -0.68 (-36.48%) | 0.21 |
| December (rural) | 3.69 | 4.32 | 2.70 | 2.98 | -0.99(-26.78%) | 0.81 |





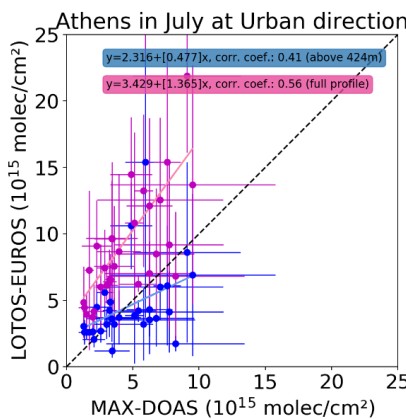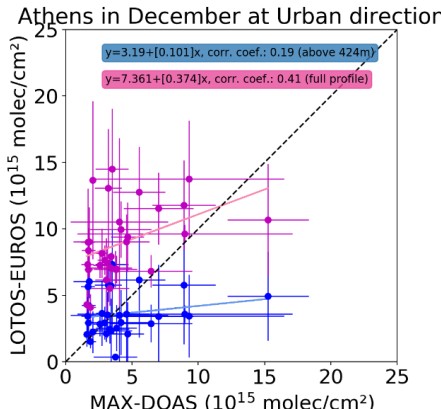

**Figure 9: Scatter plots between daily LOTOS-EUROS integrated column above 424 m (blue) and full profile (magenta) with MAX-DOAS NO₂ columns in Athens for July (left) and December 2018 (right) for the urban direction. The linear regression equation and the correlation coefficient between the model and the observed data are given in the top right of each plot.**

Figure 10 shows the average diurnal cycle of the urban direction in July (left) and December (right) of the LOTOS-EUROS full profile and partial column above 424 m, the MAX-DOAS observations and the boundary layer height used for the simulations and obtained from the ECMWF, while Figure S6 shows the average diurnal cycle for the rural direction. It is further confirmed here that the full profile column of LOTOS-EUROS captures better the daily variability of the measurements compared to the partial column (Figure 10). On the other hand the observed diurnal cycle is highly overestimated by the LOTOS-EUROS full profile while it is much similar to the partial column diurnal line. In this version of the model the mixing is more directly determined by the boundary layer height obtained from ECMWF data and possible uncertainties induced by the boundary layer height could explain the differences between the variability of the LOTOS-EUROS partial column and full profile with the measurements over Athens above the height of 500 m. We plotted the relative biases between the simulated partial column and the measurements at the urban direction in July against the boundary layer heights used. We can see that when the height of the boundary layer is relatively low (between 0 and 500 m) the model highly underestimates the measurements (Figure S7). The height of the boundary layer remains below 500 m mostly early in the morning, as seen in Figure 10, and the stronger difference between the partial column and the measurements is also observed at the same time. This could point at an underestimated boundary layer height before 7 a.m and a subsequent mixing in the model mostly in the lower heights and lower concentrations above the 500 m. At the rural direction the model's partial column underestimates the observed diurnal amplitude. The model underestimates the columns in both directions and that could be explained as well by underestimated emissions in the model or by missing transported pollution by neighbouring regions to the rural area.

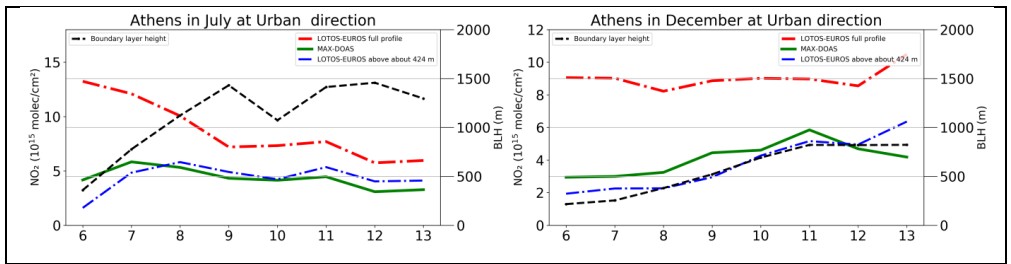

**Figure 10: Average diurnal cycle of the MAX-DOAS in Athens (green line), LOTOS-EUROS full-profile (red line) NO₂ columns, LOTOS-EUROS partial NO₂ columns (blue line) and boundary layer height (black line) during July (left) and December (right) for urban direction.**

To summarize, during July LOTOS-EUROS is well in line with the MAX-DOAS measurements over Thessaloniki with regards to the hourly measurements and the diurnal cycle of the data. In December, however, LOTOS-EUROS systematically underestimates the NO₂ columns mostly at pollution peaks. For Athens, NO₂ partial columns simulated by the model





underestimate as well the MAX-DOAS measurements in the urban and rural directions during both summer and winter months and shows stronger differences in the diurnal variability compared to the full profile of LOTOS-EUROS simulations. The boundary layer assumption appears to play a pivotal role in the case of Athens, where the NO$_2$ columns are measured above the first 500 m, and contributes to high mixing of pollutants below the 500 m early in the morning and a subsequent underestimation of NO$_2$ at higher altitudes. For both urban areas, in Thessaloniki and Athens, the model underestimates slightly

the measurements in July while the underestimation is higher during the winter month, as in the case of the surface observations, which could point at underestimated NOx emissions and too short NOx modelled lifetime. Further sources of model uncertainties include the meteorology used for the simulations, and in particular from temperature and cloudiness. In addition, the MAX-DOAS tropospheric columns in both cities have been derived using the geometric approximation without taking into account the actual NO$_2$ profile, introducing therefore, additional uncertainty. Finally, the one azimuthal directional

observation in Athens compared with a grid cell of the model may not be representative of the relatively large grid pixel of the model simulation, underestimating a possible horizontal plume from industrial areas i.e. from chimneys. Vlemmix et al., (2015) found that MAX-DOAS low daily averaged NO$_2$ columns are overestimated by LOTOS-EUROS while higher columns are underestimated, well in agreement with the results of this study at the rural directions.

**3.3 Comparison with Sentinel/5P TROPOMI vertical columns**

Sentinel/5P TROPOMI data are gridded onto LOTOS-EUROS grid with the same spatial resolution (0.1°×0.05°). The TROPOMI averaging kernels provided by the satellite product, that express the sensitivity of the instrument to the NO$_2$ abundance within the distinct layers of atmospheric column, are applied to the model profiles in order to allow a consistent comparison between the modeled and observed columns and to eliminate any possible errors in the TM5-MP a priori profile shapes (Eskes and Boersma, 2003). The averaging kernels are applied directly by the LOTOS-EUROS model. Monthly

averaged tropospheric NO$_2$ columns for the TROPOMI observations and LOTOS-EUROS simulations are given in Figure 11 and Figure 12 for July and December, respectively over the inner area of Greece and the two sub-regions of Athens and Thessaloniki. The right columns at both figures show the absolute difference between the TROPOMI and LOTOS-EUROS columns. The regions to be analysed on more details later are marked with black rectangles over the maps in Figure 11 and are named Greece, Athens and Thessaloniki as seen at the maps of the inner area, Attica Basin and Thessaloniki respectively.

Over Greece, LOTOS-EUROS captures generally well the observed NO$_2$ column abundances in July and December (upper panels of Figure 11 and Figure 12, respectively), such as the densely populated area of Athens and the lignite-burning power plants at the northwest of Greece, in the area of Ptolemaida. We can further note that the TROPOMI background NO$_2$ columns are higher than LOTOS-EUROS and this is mostly noticeable during July when the mean bias is strongly negative, at -0.59×10$^{15}$ molec.cm$^{-2}$ (-45%) and the spatial correlation coefficient is lower than in December (0.78 and 0.87 respectively)

(Table 6). The high background levels of TROPOMI can be easily distinguished in the difference plot (right column) where the purple color covering the entire region is about -0.50×10$^{15}$ molec.cm$^{-2}$. Comparison of the TROPOMI NO$_2$ data with Pandora measurements in Helsinki showed that TROPOMI slightly overestimates the NO$_2$ columns when they are relatively low, and underestimates the high columns (Ialongo et al., 2020). This is further shown by Dimitropoulou et al. (2020) who validated TROPOMI NO$_2$ tropospheric columns with MAX-DOAS measurements over an urban area in Belgium and

confirmed that TROPOMI underestimates the measurements by about 40-50% at urban sites and argued the need of more appropriate a priori profiles in the TROPOMI algorithm retrieval. However, since in our case the averaging kernel is applied there is no NO retrieval profile-related bias influencing the comparisons. Therefore, the large background difference between the two datasets may be as well a result of background column missing in the model simulations. This might be related to an underestimation of the free tropospheric column by the model due to missing lightning emissions. Secondly, the profiles of

LOTOS-EUROS peak more strongly near the surface, and this leads to a smaller model-simulated TROPOMI value and a subsequent strong difference in the free troposphere. The slope for July is 0.40 and the offset 0.22×10$^{15}$ molec.cm$^{-2}$ as seen in

Table 6. Moreover, in December the LOTOS-EUROS NO$_2$ columns are slightly higher over the high polluted areas such as the northwest of Greece and the temporal correlation between TROPOMI and LOTOS-EUROS is lower than in summer (0.32 and 0.44 respectively) unlike the spatial correlation that is higher in winter, and the bias is very low and slightly positive at 0.09×10$^{15}$ molec.cm$^{-2}$ (5.9%) showing a small model overestimation. The slope in this case is equal to 0.30 while the offset is 1.15×10$^{15}$ molec.cm$^{-2}$ (Table 6).

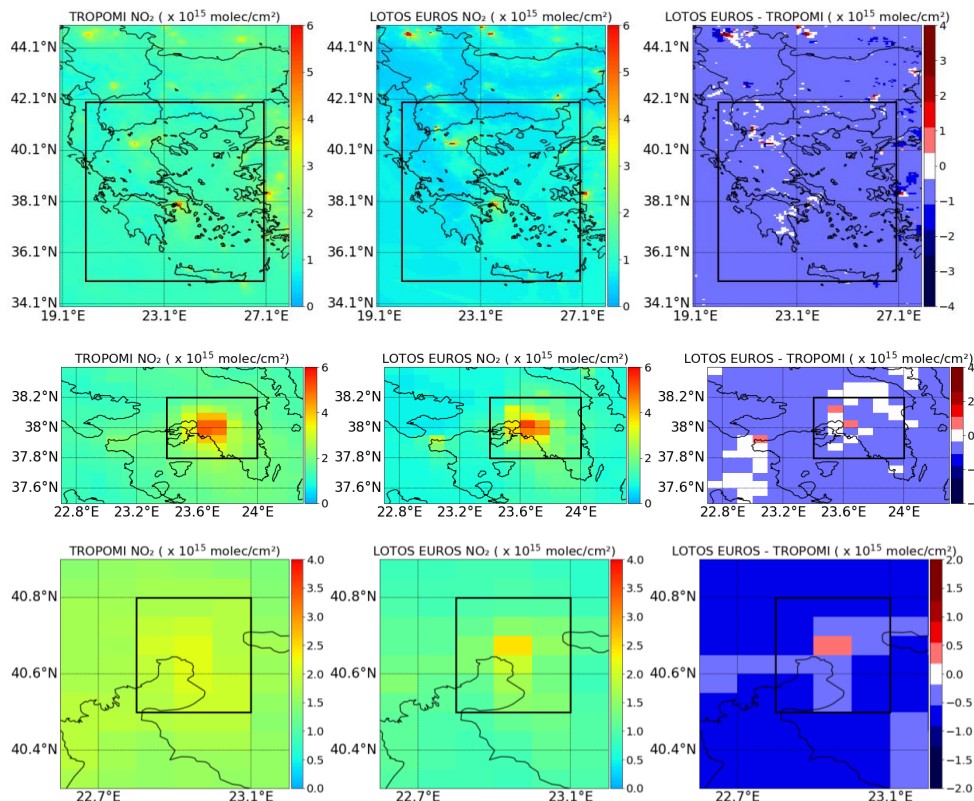

**Figure 11: TROPOMI (left column) and LOTOS-EUROS (middle column) NO$_2$ vertical columns over the region of Greece (upper panel) and the sub-regions of Athens (middle panel) and Thessaloniki (lower panel) for July and their absolute differences (right column).**

Statistics and maps over the sub-regions of the two largest and most populated regions in Greece, i.e. Athens and Thessaloniki, are given along with the ones in Greece to draw further conclusions. TROPOMI shows higher NO$_2$ columns than LOTOS-EUROS in July at the sub-region of Athens (Figure 11, middle panels). The spatial correlation when we study alone the polluted region of Athens is very high and equal to 0.95 while the temporal correlation is 0.48 and the bias quite low and still negative, about -17.9% (-0.48×10$^{15}$ molec.cm$^{-2}$,) as seen in Table 6. In December, the NO$_2$ columns of LOTOS-EUROS are higher mostly at the southern part of the sub-region of Athens comparing to the TROPOMI (Figure 12, middle panels), while the spatial correlation between them is lower than in summer (0.82) and the temporal correlation much higher than in summer (0.84). The bias is positive in this case 0.74×10$^{15}$ molec.cm$^{-2}$ (16.8%) showing again an overestimation of the model during the winter month comparing with the TROPOMI observations. Lastly, when comparing the TROPOMI and LOTOS-EUROS NO$_2$ columns over Athens in both July and December the components of the linear regression applied are good. The slopes are 0.49 and 0.85 and the intercepts are 0.22×10$^{15}$ molec.cm$^{-2}$ and 1.46×10$^{15}$ molec.cm$^{-2}$ in July and December respectively.



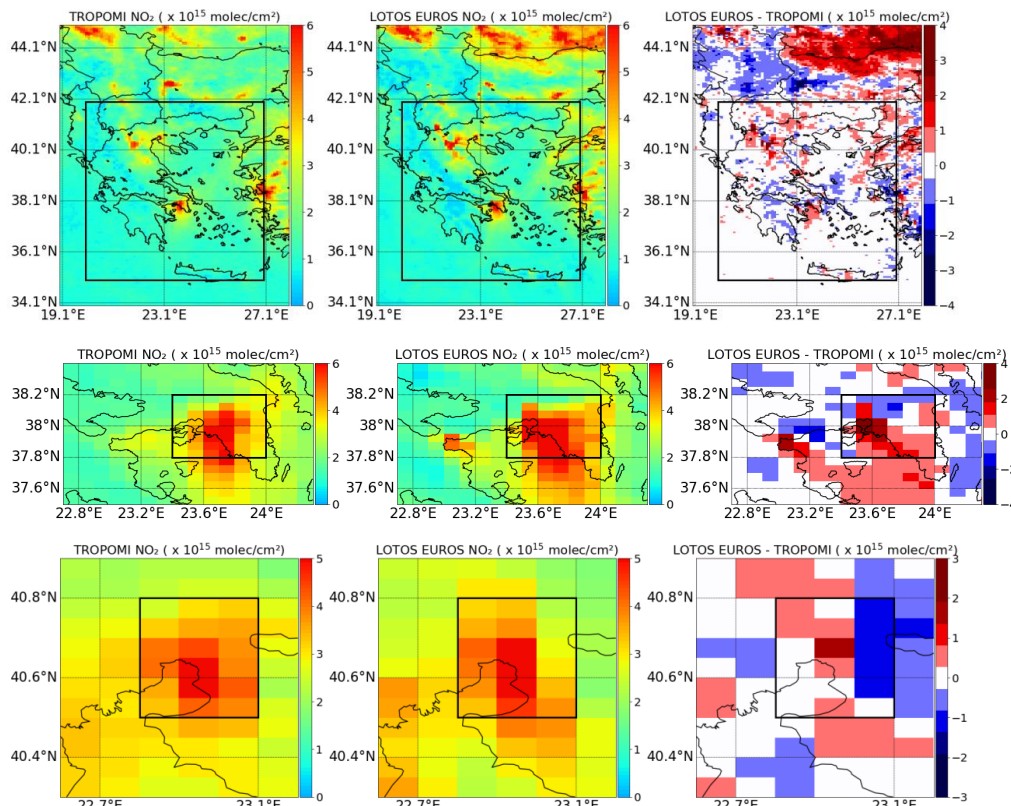

**Figure 12: TROPOMI [left column] and LOTOS EUROS [middle column] NO₂ tropospheric vertical columns over the region of Greece [upper row] and the sub-regions of Athens [middle row] and Thessaloniki [lower row] for December and their absolute differences (right column).**

Thessaloniki shows a similar behaviour as Athens (lower panels of Figure 11 and Figure 12), with higher NO₂ columns observed by TROPOMI in July and lower in December in comparison with LOTOS-EUROS simulations. In July, their spatial correlation is higher than in December (0.82 and 0.66 respectively) while the temporal correlation is lower than in the winter month (0.30 and 0.58 respectively)(Table 6). LOTOS-EUROS underestimates the NO₂ column during July (the bias is -0.52×10¹⁵ molec.cm⁻²) and overestimates slightly in December (the bias is 0.15×10¹⁵ molec.cm⁻²) as well in Thessaloniki.

The LOTOS-EUROS results over Athens in Figure 11 and Figure 12 clearly show a peak at the Isthmus of Corinth, the narrow land bridge which connects the Peloponnese peninsula with the rest of the mainland of Greece, near the city of Corinth. Corinth has an important port (mostly cargo), while vessels navigate through the canal, and it is also an industrial area home to the largest oil refining company in Greece. In December (Figure 12, middle panels) LOTOS-EUROS simulates high NO₂ columns (mean value ~5×10¹⁵ molec.cm²) near the Isthmus of Corinth, which are not supported by the TROPOMI observations, pointing

to a possible overestimation of the NOx emissions in the area. Possible NOx reductions in the area should be studied when emission inventories for year 2018 will be released.

**Table 6 Statistics between the LOTOS-EUROS vertical NO₂ columns and the TROPOMI observations for July and December 2018. Columns are expressed in 10¹⁵ molec.cm⁻²**

|  | TROPOMI | LOTOS-EUROS | Bias (%) | Temporal Corr. | Spatial Corr. | Slope (spatial) | Offset (spatial) |
|---|---|---|---|---|---|---|---|





| | Mean | STD | Mean | STD | | | | | |
|---|---|---|---|---|---|---|---|---|---|
| Thessaloniki (July) | 1.92 | 0.48 | 1.40 | 0.77 | -0.52 (-26.9%) | 0.30 | 0.82 | 0.33 (2.08) | 0.78 (-2.59) |
| Athens (July) | 2.70 | 1.65 | 2.22 | 2.05 | -0.48 (-17.9%) | 0.48 | 0.95 | 0.49 (1.07) | 0.96 (-0.69) |
| Greece (July) | 1.31 | 0.48 | 0.72 | 0.42 | -0.59 (-45%) | 0.44 | 0.78 | 0.40 (0.82) | 0.22 (-0.35) |
| Thessaloniki (December) | 3.88 | 2.43 | 4.04 | 2.85 | 0.15 (3.6%) | 0.58 | 0.66 | 0.56 (1.00) | 2.05 (0.12) |
| Athens (December) | 4.02 | 2.93 | 4.77 | 4.25 | 0.74 (16.8%) | 0.84 | 0.82 | 0.85 (1.25) | 1.46 (-0.34) |
| Greece (December) | 1.43 | 1.09 | 1.52 | 1.46 | 0.09 (5.9%) | 0.32 | 0.87 | 0.30 (1.14) | 1.15 (-0.12) |

Time series for the mean columnar levels of the three domains studied (i.e. Greece, Athens and Thessaloniki) for July and December are shown in Figure 13. For the region of Greece (upper panel) in July (left) we can see that the model reports low and nearly invariant $NO_2$ columns while TROPOMI shows slightly larger daily variability, while in December (right) they both show similar daily patterns. Of course there are days on which the simulations agree well with the observations (18/12-21/12 and 29/12-31/12) while on others, they deviate (25/12 and 26/12). For Athens (middle panel) and Thessaloniki (lower

panel) we can see that the simulated columns are variable; in July the CTM underestimates on average the satellite observed $NO_2$ columns for both regions, while in December, they follow closely the high values of TROPOMI which explains the higher temporal correlations in winter.

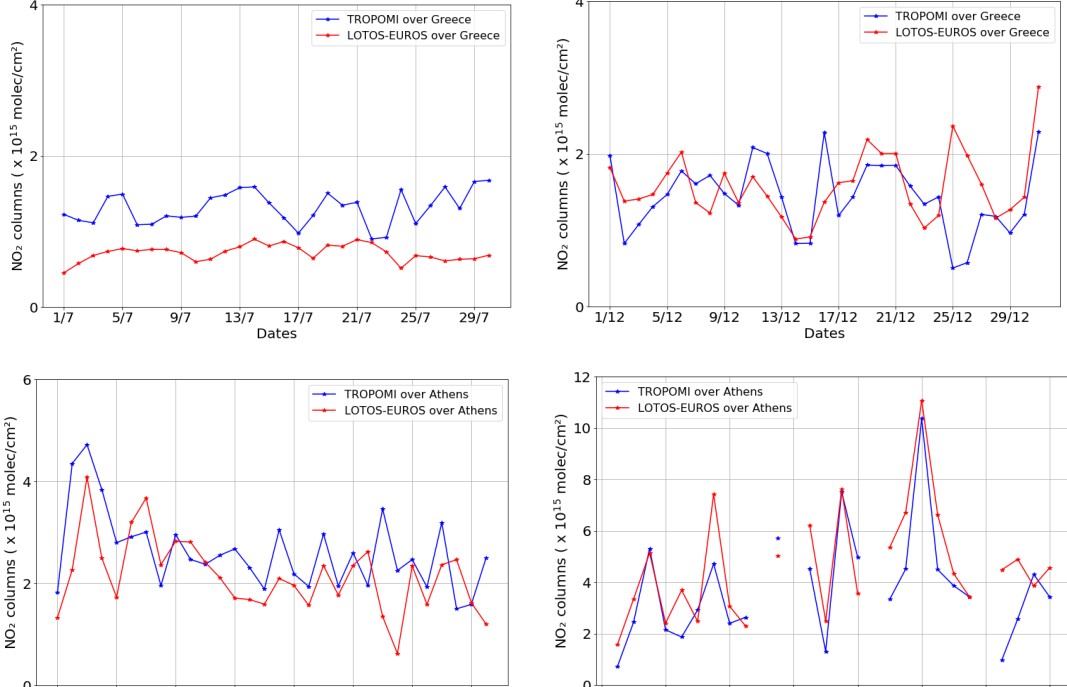





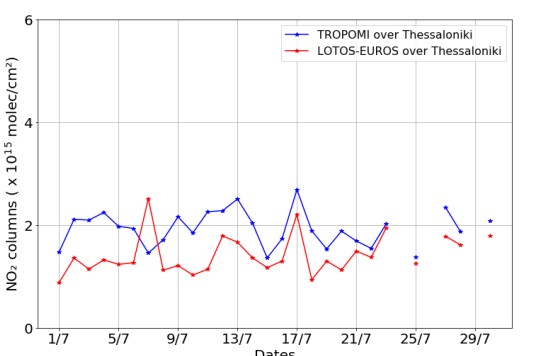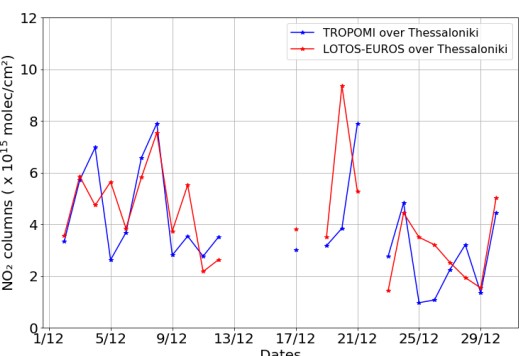

**Figure 13 Time series of average TROPOMI (blue lines) and LOTOS-EUROS (red lines) NO₂ columns over the 3 distinct domains at the TROPOMI overpasses: the domain over Greece (top), the domain over Athens (middle) and the domain over Thessaloniki (bottom). July is shown in the left column and December in the right.**

To conclude, LOTOS-EUROS, in general, underestimates the NO$_2$ columns during July over Greece compared to the TROPOMI observations while it overestimates the NO$_2$ levels during December. The emission inventory used as input for the simulations is outdated by a few years and this fact may impact on the NO$_2$ simulations. Verhoelst et al., 2020 reported that TROPOMI shows an underestimation of NO$_2$ tropospheric columns of -23 to -37% in clean to slightly polluted conditions that corroborates previous findings such as those of Ialongo et al. (2020) and could partly explain the model overestimation in December at the two urban centres. However the simulations are well in line with the observations vis-a-vis the spatial distribution of the NO$_2$ values, and mostly over the region of Athens. For both Thessaloniki and Athens in summer the spatial correlation is higher than in winter while the temporal correlation is lower. The model underestimation of the background values in summer compared to satellite retrievals according to Huijnen et al. (2010) can be a result of an underestimation of NOx lifetime in the model as well as an under-prediction of the transport processes in the free troposphere. NOx emissions from lightening also are not included in the model and as a result some NO$_2$ is missing in the free troposphere. While it is generally assumed that underestimated or missing soil-NO emissions usually form an important factor of uncertainty in the models note that the background values of TROPOMI remain high even over the sea, indicating that natural emissions might not be the cause of the high background values. Additionally, the simulations are considered at the TROPOMI overpasses over the Greek domain, about 12 p.m., a time when LOTOS-EUROS has already demonstrated low NO$_2$ surface concentrations when compared with in-situ measurements. In summer the bias between the surface NO$_2$ measured by the in-situ stations in the region of Thessaloniki is -16.2% while in Athens is -3.8% and the biases between the TROPOMI and LOTOS-EUROS columns for the same regions are -26.9% and -17.9% in July respectively (Table 7). However, in winter the model underestimates as well the surface observations in Thessaloniki and Athens (-33.1% and-15.9% respectively) while overestimates the TROPOMI tropospheric columns (3.6% and 16.8% respectively). For the same grid cells that we analyzed the MAX-DOAS observations in Thessaloniki and Athens we extracted respectively the TROPOMI observations for July and December and in-situ measurements when available for the summer and winter periods and we show the models biases (Table 7). For the grid cell in Thessaloniki (AUTH) we found that the model underestimates all kind of observations in July with a higher negative bias found for the surface simulations (-20.8%), while in winter LOTOS-EUROS highly underestimates both the surface observations and the tropospheric column measured by MAX-DOAS (-39.2% and -32.8% respectively) and slightly overestimates the TROPOMI observation (8%). In the urban grid cell in Athens we found the same characteristics as in AUTH, where in summer the model underestimates the surface measurements and columnar observations while in winter as well only the TROPOMI observations are overestimated (8.6%). The MAX-DOAS and the in-situ measurements are underestimated only by 2%-3% while TROPOMI shows a higher negative underestimation (-12.6%) in July. The rural grid cell measurements of NO$_2$ tropospheric columns are underestimated by the model when compared to MAX-DOAS and TROPOMI observations both in summer and in winter.





**Table 7 Summarized table of relative biases between LOTOS-EUROS NO₂ surface simulations and in-situ measurements and between simulated and observed vertical tropospheric columns. The "grid cell" in the "Region" column refers to biases of NO₂ values found at the same grid cell, while "mean area" refers to the biases calculated for the total area in summer and winter. The positive biases are shown in bold.**

| Region | Measurement | Summer | Winter |
|---|---|---|---|
| AUTH/grid cell | S5P/TROPOMI | -11.4% | **8.0%** |
| | MAX-DOAS | -8.4% | -32.8% |
| | In-situ | -20.8% | -39.2% |
| Thessaloniki/mean area | S5P/TROPOMI | -26.9% | **3.6%** |
| | In-situ | -16.2% | -33.1% |
| Athens Urban/ grid cell | S5P/TROPOMI | -12.6% | **8.6%** |
| | MAX-DOAS | -2.2% | -14.5% |
| | In-situ | -3.0% | -17.0% |
| Athens Rural/ grid cell | S5P/TROPOMI | -38.2% | -13.4% |
| | MAX-DOAS | -36.5% | -27.8% |
| Athens/mean area | S5P/TROPOMI | -17.9% | **16.8%** |
| | In-situ | -3.8% | -15.9% |

## 4 Conclusions

In this work, we evaluate NO₂ simulations over Greece from the LOTOS-EUROS regional CTM using in-situ surface concentrations from 14 air quality stations during June and December 2018. Further we compare LOTOS-EUROS columns against MAX-DOAS and Sentinel/5P TROPOMI tropospheric columns in July and December 2018. The model setup is based on the anthropogenic emission inventory TNO-CAMS v2.2 for the year 2015, ECMWF meteorological data and CAMS near-real time initial and boundary conditions. We conclude the following:

Overall, the spatial correlation between the modelled and the measured surface NO₂ values ranges between 0.70 and 0.86 depending on the time period. The mean temporal correlation coefficient is equal to 0.52 and the mean bias is -10% from June to December period. LOTOS-EUROS follows nicely the hourly variability of the measurements during daytime (12 to 15 p.m. local time) and the temporal correlations between the simulations and the measurements range between 0.43 and 0.72 (excluding the traffic station "Piraeus" due to the impact of high traffic emissions close to it), while overall underestimates the measurements over the 14 stations in Greece. During night-time (0 to 3 a.m. local time) the model overestimates the surface NO₂ when the measurements are low suggesting a low boundary layer height assumption. A slight dependency on the season was found since the model underestimates stronger the NO₂ values during winter showing an average relative bias of -15% and a spatial correlation coefficient of 0.78, while in summer the average relative bias is -1% and the spatial correlation coefficient reaches 0.86. The mean temporal correlation coefficient in both seasons is similar (about 0.50).

Very good agreement in the diurnal cycle variability of the LOTOS-EUROS NO₂ columns compared to those of the MAX-DOAS instrument in Thessaloniki is found for July, as well as a small underestimation (relative bias of -8.44%) of the observations. The underestimation of the measurements is higher during December (-32.8%). The model underestimates as well the NO₂ columns over the urban region of Athens negligibly in July (-2.2%) and slightly stronger in December (-14.5%), while follows much better the diurnal cycle when the model's full profile is considered compared to the partial column above the 424 m, pointing to uncertainties in the boundary layer height which could depend on parameters that affect urban environments too and are not taken into account (e.g. the urban heat island). The measurements at the rural direction are



underestimated in both summer and winter showing possible underestimation of pollution transfer from neighboring regions, while in winter the correlation is pretty high and equal to 0.81.

The model reproduces very well the spatial variability of TROPOMI $NO_2$ columns over Greece capturing the locations of low and high $NO_2$ columns. The spatial correlation between the simulations over Athens and the TROPOMI observations is 0.95 in July and 0.82 in December while the levels of $NO_2$ are underestimated and overestimated respectively in summer and winter

by ~18%. The same characteristics are observed over the city of Thessaloniki as well, with higher spatial correlation in summer (0.82) and negative relative bias (-26.9%) and lower spatial correlation in winter (0.66) and a negligible positive bias (3.6%). Higher background values of $NO_2$ are observed in TROPOMI product mainly during summertime possibly due to an underestimation of the free tropospheric column (missing lightening emissions in the simulations, under-prediction of the transport processes in the free troposphere) and the model-simulated TROPOMI column (higher concentrations of LOTOS-

EUROS near the surface).

The underestimation of the MAX-DOAS columns in summer and winter at the urban area of Thessaloniki is consistent with the negative biases of the surface observations and could point at underestimated emissions or under predicted lifetime of $NO_2$ mostly during the winter season. Same characteristics are found for the urban region in Athens, where MAX-DOAS views the troposphere above the first about 500 m, suggesting lower $NO_2$ concentrations at higher altitudes as well. The relative biases

between TROPOMI and the modelled columns for both areas of Athens and Thessaloniki are negative and higher than in the case of surface measurements possibly due to missing emissions at higher altitudes (i.e. lightening). However, in winter the surface measurements are underestimated by the model and the vertical TROPOMI columns overestimated possibly due to the already known TROPOMI $NO_2$ underestimation over slightly polluted areas. Further studies on emission inversions using the data assimilation package of LOTOS-EUROS should be conducted in a next step to account for uncertainties due to outdated emission inventories. Further, the model simulations found to rely on possible uncertainties on the meteorological input data

and improvements when the height of boundary layer is very low or very high are suggested.

*Author Contributions*: The satellite and CTM data analysis was performed by I.S.; methodology and conceptualization by M.E.K. and D.B.; software development by I.S.; MAX-DOAS observations were performed by D.K. and M.G; writing— original draft preparation by I.S. and M.E.K.; review and editing by all authors. All authors have read and agreed to the

published version of the manuscript.

*Acknowledgments:* We acknowledge the usage of modified Copernicus Sentinel data 2019-2020. Results presented in this work have been produced using the Aristotle University of Thessaloniki (AUTh) High Performance Computing Infrastructure and Resources. M.E.K., I.S and D.B. would like to acknowledge the support provided by the IT Center of the AUTh throughout the progress of this research work.

*Financial support*: This research has been co-financed by the European Union (European Regional Development Fund) and Greek national funds through the Operational Program "Competitiveness, Entrepreneurship and Innovation" (NSRF 2014-2020) by the "Panhellenic Infrastructure for Atmospheric Composition and Climate Change" project (MIS 5021516) and well as the "Innovative system for Air Quality Monitoring and Forecasting" project [code T1EDK-01697, MIS 5031298), implemented under the Action "Reinforcement of the Research and Innovation" Infrastructure.

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
