# Peer review of "Evaluation of the LOTOS-EUROS NO2 simulations using groundbased measurements and S5P/TROPOMI observations over Greece"

_Atmospheric Chemistry and Physics, 2020_

## Referee Comment (RC1) · Anonymous Referee #1 · 26 Nov 2020

Review on the manuscript: "Evaluation of the LOTUS-EUROS NO2 simulations using ground-based measurements and S5P/TROPOMI observations over Greece" by Skoulidou et al.

The manuscript describes the comparison between LOTUS-EUROS NO2 model simulations and in situ, max-doas and TROPOMI data. The comparison shows that the model reproduces well the spatial variability of in situ measurements and TROPOMI NO2 observations. Overall the agreement changes with the season and at different sites. The paper is scientifically sound, but it is missing deeper analysis of the uncertainties and it is quite busy. The manuscript could be published after addressing the following points:

Specific comments:

1) As a general point the paper is not enough focused in my opinion. The results are a bit scattered (presented for many instruments and conditions) and it is difficult to derive a clear take home message. Perhaps the authors could try to rewrite more clearly the abstract and conclusions (now they are just summaries), for example highlighting under which conditions the model performs best and worst and the main reasons for discrepancies and ideas for improvement. Most of this is perhaps already indirectly mentioned in the text, but I think it could be rewritten in a different manner, so that the model capabilities and limitations can be better highlighted.

2) sect. 2.2.1 you mention that "For this reason, stations characterized as urban traffic stations, localised close to busy traffic roads of the city and showing very large values, are excluded from the validation." But then you analyse some urban traffic stations in the results. Which criteria you used to exclude these stations? Also, I think you will still have differences in spatial representativeness, also when NO2 values are not "very large": please clarify.

3) It would be useful to plot the actual grid of the model for the Greek (nested) domain (0.1x0.05 deg ) in figure 1 and 2. This would show that actually sometimes more than one in situ station fall into one grid cell of the model (at least in Thessaloniki). Did you try (if applicable) to average spatially the values from the stations within one grid cell and see if it reduces some of the discrepancy in the representativeness between model and point measurements?

4) The discussion on the uncertainty is a bit qualitative sometimes. Many figures lack errorbars (see technical points below). For example, what are the uncertainties associated with individual max-doas measurements? If you use an average over time, could you include some estimation of the variability in this time range? Also, how much

do you expect your geometric AMF calculation to change the result compared to the calculation that takes into account the actual NO2 profile? The same applies for the comparison with TROPOMI NO2. A more quantitative description of the uncertainties would also help in understanding how the discrepancies you find compare with these uncertainties.

5) Sect. 3.2 Maybe I lost this information but which direction you use for AUTH: 1 or 2?

6) Sect. 3.3 Could you please clarify how do you apply the averaging kernels of TROPOMI to the model? How do you spatially and temporally collocate TROPOMI and the model? How do you interpolate vertically? Which level you use for the tropopause (from TROPOMI perhaps)?

Technical comments

7) Table 7 you write in the caption: "The positive biases are shown in bold." But there is no bolded text in the table. Also, one horizonal line is missing.

8) Figure 7. The y-axis title of the second panel is not visible here

9) Figure 7 and 10 should have errorbars.

10) Figure 9. Please write in the caption what the errorbars are.

11) Overall, the paper is a bit figure and table -heavy in my opinion and a bit repetitive sometimes. Maybe you can try to shorten some text throughout. For example, while it is useful to have these summaries at the end of each paragraph, it could be written in a more concise manner. Also, some of the tables and figures could go to the supplement. For example, Fig. 4, Fig. 13, Table 7 could be moved to the supplement. Perhaps some figures could be also grouped together.

---

## Referee Comment (RC2) · Anonymous Referee #2 · 1 Dec 2020

The paper "Evaluation of the LOTOS-EUROS NO2 simulations using ground-based measurements and S5P/TROPOMI observations over Greece" by Skoulidou et al., presents LOTUS-EUROS NO2 simulation over Greece, for a period of 7 months, from June to December 2018. The data are compared to in-situ NO2 concentrations for the whole period at 14 sites in Athens and Thessaloniki, then to 2 months (June and December) of tropospheric NO2 VCD from MAX-DOAS instruments in Athens and Thessaloniki and from TROPOMI. Differences as a function of the season are discussed. The scientific content of the paper fits the scope of ACP, and the paper is interesting,

although a bit difficult to read due to length and repetition in several sections. The number of tables and figures could also be reduced. It is a pity that profiles are not exploited a bit more. It would be nice to have: 1) comparisons of the LOTOS-EUROS NO2 profiles wrt to TROPOMI a-priori profiles from TM5, and 2) at the MAX-DOAS stations, profiles retrievals (instead of only tropospheric VCD from geometrical approximation) could be used as a link between the surface NO2 from the in-situ and the tropospheric VCD from TROPOMI. Moreover, the LOTUS-EUROS profiles could be used to test a relation between surface and tropospheric VCD, and test this assumption with the in-situ and MAX-DOAS measurements. I would thus recommend some revision of the text and some further investigations, as described below.

specific comments:
* * *
- why not compare also to MAX-DOAS from June "to" December (as for the in-situ) instead of June "and" December?

- Are MAX-DOAS profiles available? they could make the link between surface NO2 values from in-situ and tropospheric VCD

- Also how are the LOTOS-EUROS NO2 profiles comparing to TROPOMI a-priori profiles from TM5?

- how are the TROPOMI AVK applied to the LOTUS-EUROS model? It is said that gridded data are created from TROPOMI pixels. Are the AVK averaged to created a grid of AVK?

P4, L 114: there are 10 levels "from the surface to a top around 175 hPa (about 12 km)." Are all the levels of same width?

P4, L 150: "For this reason, stations characterized as urban traffic stations, localised close to busy traffic roads of the city and showing very large values, are excluded from the validation." –> how this selection is done? we still have urban traffic sites in Sect

3.1...

P5, L.160: why only "for July and December" and not between July and December as for the in-situ comparisons (or over the whole year)?

P5, L.165: why the 15° elevation has been chosen and not the 30° elevation? is there any further selection, as in Drosoglou et al., 2017 where an average of both elevations was considered if the results from the 2 angles are within 20 or 30% ?

P5, L.166: both the azimuth 220° and 255° are looking over the sea. Do you have viewing directions in the same direction than the in-situ measurements (from AUTH to Lagada and Kordelio (this last name and Sindos are difficult to read in Figure 2))?

P5, L.181: for Athens, the geometrical approximation is also used, but from which elevation angle? also 15°, as for Thessaloniki?

P5: please specify if MAX-DOAS data are filtered for clouds, and give an estimation of the errors on the tropospheric VCD for both sites. Are they of equivalent quality? Please also discuss the MAX-DOAS horizontal representativeness area (or at least mention the outcome from Drosoglou et al., 2017 and Gratsea et al., 2016). Are these taken into account in the comparison, or is the MAX-DOAS considered as a "point measurement" in the horizontal plane?

P6, Fig2: in this figure, several in-situ stations are in the line of sight of one of the MAX-DOAS azimuth direction. Do they show similar diurnal variation? or can these be compared using the model profile shape to convert surface to tropospheric VCD? Similarly, adding the extension of the LOTOS-EUROS 0.1°×0.05° grid on these maps, could help the reader understanding if several in-situ stations are in one model grid cell.

P6, L..203: "that because the averaging kernels are used in our case, the comparison with LOTOS-EUROS is not influenced by the retrieval a-priori (Eskes and Boersma, 2003" –> this is the case for LOTOS-EUROS vs TROPOMI, but not eg for the LOTOS-

EUROS vs MAX-DOAS comparisons. Moreover, this is a bit misleading, as the coarse apriori model profiles would still play a role. Are the TROPOMI AVK also gridded "onto the LOTOS-EUROS grid at 0.1°×0.05°." (P.6, L 208)? How are the AVK applyed? Please explain.

P7, Sect 3.1: what is the width of the first model level, that is compared to the in-situ surface concentrations? The discussion would be more easy to follow if instead of the tables 1 and 2 (or in addition, maybe in the supplement) a few plots of the diurnal variation of the in-situ and the model at the sites is shown (a bit like figures 7 and 10). It would be more easy to also understand why the 12-15pm is selected as representative of "daytime" conditions and 0-3am of "night" condition. Are there big changes outside these periods? It would also allow to draw a conclusion on the consistency (or not) of the diurnal surface NO2 variation compared to the diurnal NO2 VCD variation.

Figure 7 and 10: what would be the MA X-DOAS if retrieved from 30° elevation instead of 15°?

P20, Sect4: the main messages are a bit lost in the conclusion, which is a bit too much a repetition of each subsection conclusions.

Technical comments and corrections:

——————————————————–

P2, L.71: (Zerefos et al., 2000.) –> (Zerefos et al., 2000).

P4, L.117: what is the "tree-species database"?

P4, L.135: 2.2Ground-based –> add a space after the section number

P4, L.143: give some references for the chemiluminescence method

P5, L.172: quantify "good agreement"

P6, L.194: "several TROPOMI NO2 validation papers that have been recently submitted (Judd et al., 2020; Verhoelst et al., 2020)." –> there are some TROPOMI NO2 validation papers already fully published: Zhao et al 2019 (https://www.atmos-meas-techdiscuss.780net/amt-2019-416/ ), Ialongo et al., 2020 ( https://doi.org/10.5194/amt-13-205-2020 )

P7, L.229: "the correlations are very good (r=0.69 and 0.63" –> remove "very"

P7, L.234: "In this case, a clear seasonal pattern in the model's performance, as is the case for Thessaloniki, was not found." –> do you have any hint why?

P7, Table 1 and 2: please add the units of the RMSE.

P10, L.283: please add in Figure S3 caption or ylabel, the definition of relative biases. (simulation-obs)/obs? Also, it could be nice to have a different symbol for each site, so that it would be clear for the reader which site(s) are the outliers of the whiskers in Winter and night conditions. Either 14 symbols, either grouped by station types introduced in tables 1 and 2, either one color per Athens, one per Thessaloniki...

P10, L.394: please specify how "spatial correlation coefficient" and "temporal correlation" are calculated.

P10, L.298: "representation issues related to the location of the stations" –> link with the Drosoglou et al., 2017 study with high resolution model (6km resolution for the Balkans and 2km resolution for the Thessaloniki region)

P11, L/ 320: "The MAX-DOAS in the center of Thessaloniki observes high NO2 columns during the winter months and lower levels during the spring season..." –> is this a description of the rest of the MAX-DOAS dataset, not shown here, or is "spring season" mis-referring to July data or to the Drosoglou 2017 results?

If other months than December and July are available from the MAX-DOAS, how are they comparing to LOTUS-EUROS?

P11, L.237: "The daily mean" - how are the daily mean performed? is it , as for the

in-situ, only 12 to 15pm, or is it all the available points (below 75°SZA)? Is there a difference between the 2 approaches for MAX-DOAS data?

P12, L.355: are the MAX-DOAS data cloud filtered? if there are some gaps in the MAX-DOAS, are these gaps considered also in the model data, before doing the daily average?

Figure 7 and 10: please increase a bit the size of these figures. The legend is diffcult to read.

P15, L.423: "the MAX-DOAS tropospheric columns in both cities have been derived using the geometric approximation without taking into account the actual NO2 profile, introducing therefore, additional uncertainty" –> please estimate this error.

P15, L424: "the one azimuthal directional observation in Athens compared with a grid cell of the model may not be representative of the relatively large grid pixel of the model simulation, underestimating a possible horizontal plume from industrial areas i.e. from chimneys" –> mention and discuss a bit more the MAX-DOAS horizontal representativity and the model size.

P15, L.434: "The averaging kernels are applied directly by the LOTOS-EUROS model" –> "by" or "to" ? Explain better how the AVK are applied (gridded AVK? application of AVKK at the pixel level, and then gridding? ...?)

P15, L.447: Pandora measurements in Helsinki are total columns!

P15, L 452: "there is no NO retrieval profile-related bias influencing the comparisons" –> NO to NO2 this is partially true, but the influence of the coarser TM5 1x1 degree resolution instead of a regional high resolution model is still present (see Zhao et al., 2019).

P15, L.455: "the profiles of LOTOS-EUROS peak more strongly near the surface" –> it would be interesting to see the comparison of the profiles shapes (TM5 vs LOTOS-EUROS).

P17, L.488: "In December (Figure 12, middle panels) LOTOS-EUROS simulates high NO2 columns (mean value ∼5×1015 molec.cm2) near the Isthmus of Corinth, which are not supported by the TROPOMI observations, pointing to a possible overestimation of the NOx emissions in the area" - it could maybe also be related to winds that do not add up? It would be nice to see the TROPOMI December map if the winds speed and direction would be taken into account to create the map (Zhao et al., 2019; Lorente et al., 2019)

Lorente et al., 2019: https://www.nature.com/articles/s41598-019-56428-5

Zhao et al 2019: https://www.atmos-meas-tech-discuss.780net/amt-2019-416/

Ialongo et al., 2020: https://doi.org/10.5194/amt-13-205-2020

---

## Referee Comment (RC3) · Anonymous Referee #3 · 14 Dec 2020

The authors compare the LOTOS-EUROS simulations of NO2 over Greece against surface measurements, DOAS profiles and sentinel maps during the second half of 2018. The comparison is also performed at different seasons, sites and hour of the day, and the authors provided reasoning for the differences. The paper is within the scope of the journal and it is scientifically sound. My main concern is on the significance of some results, which affects its emphasis and extent. I trust it should be published, following the recommendations hereafter.

Specific Comments 1. The validation approach relies mainly on correlation and RMSE.

[Figure]

The linear correlations should be tested for their significance. The same applies also for the spatial correlations, for which, the estimation algorithm is missing. Use of NRMSE is more informative when comparing the simulations at different stations. 2. Can the authors comment on the impact of the 24h periodicity to the temporal correlations? 3. The way and reason some stations have been excluded should be re-framed to be less qualitative. 4. The comparison of the gridded LOTOS-EUROS simulations against point measurements needs some clarifications. Ideally, one should either compare the observations with the simulations pin-pointed at the station location or the model grid values with the cluster of observations falling inside. 5. Uncertainty estimates require a more rigorous framework, with a better description. 6. The comparison of the gridded LOTOS-EUROS simulations against satellite data needs some clarifications on the TROPOMI data regridding and the application of the averaging kernel in LOTOS-EUROS.

Technical Comments Tables: Please specify which correlations are significant. Figures: The information in some figures is not easily seen (e.g. Figure 4, 5).

---

## Author Comment (AC1) · 8 Feb 2021

Review on the manuscript: "Evaluation of the LOTOS-EUROS NO2 simulations using ground-based measurements and S5P/TROPOMI observations over Greece" by Skoulidou et al. The manuscript describes the comparison between LOTUS-EUROS NO2 model simulations and in situ, max-doas and TROPOMI data. The comparison shows that the model reproduces well the spatial variability of in situ measurements and TROPOMI NO2

observations. Overall the agreement changes with the season and at different sites. The paper is scientifically sound, but it is missing deeper analysis of the uncer-
tainties and it is quite busy. The manuscript could be published after addressing the following points:

Specific comments:

[Figure]

1)	As a general point the paper is not enough focused in my opinion. The results are a bit scattered (presented for many instruments and conditions) and it is difficult to derive a clear take home message. Perhaps the authors could try to rewrite more clearly the abstract and conclusions (now they are just summaries), for example highlighting under which conditions the model performs best and worst and the main reasons for discrepancies and ideas for improvement. Most of this is perhaps already indirectly mentioned in the text, but I think it could be rewritten in a different manner, so that the model capabilities and limitations can be better highlighted.

Following the reviewer's suggestions changes in the structure and the writing of the manuscript are made in order to make the take home message clearer.

New sections were added in the section 3, where results are shown, to discuss the capabilities and limitations in each comparison section. As a result, the sections added are:

 3.1.1    Discussion on the validation of surface NO$_2$ concentrations

 3.2.3    Discussion on the validation of tropospheric NO$_2$ columns against ground-based MAX-DOAS observations

3.3.1    Discussion on the validation of tropospheric NO$_2$ columns against S5P/TROPOMI observations

Moreover the conclusions are changed to Conclusion and Discussion and are rewritten in order to make the take home message clearer.

[Figure]

Finally some Figures and Tables are moved to the supplement or removed completely (i.e. Table 3, Figure 9) in order to make the paper less busy.

2)      sect. 2.2.1 you mention that "For this reason, stations characterized as urban traffic stations, localised close to busy traffic roads of the city and showing very large values, are excluded from the validation." But then you analyse some urban traffic stations in the results. Which criteria you used to exclude these stations? Also, I think you will still have differences in spatial representativeness, also when $NO_2$ values are not "very large": please clarify.

The official designation of the station type was assumed to be that one reported in the official databases, however due to our detailed knowledge of where those stations are located we decided to exclude ones that are exactly over busy thoroughfares in Athens. It follows that those stations are directly affected by the smallest changes in road emissions and their reported measurements far noisier than stations that are within the city canopy but not directly on a busy road. We have included the following in the text:

"For this reason, stations characterized as urban traffic stations, localised close to busy traffic roads of the city and showing extremely high concentrations were excluded from the validation, based on local knowledge of their actual locations. As a result, we include in our analysis stations that are officially characterized as "traffic stations" (e.g. Marousi station, Athens) but which are not placed directly over the major thoroughfares."

3)      It would be useful to plot the actual grid of the model for the Greek (nested) domain

(0.1x0.05 deg ) in figure 1 and 2. This would show that actually sometimes more than one in situ station fall into one grid cell of the model (at least in Thessaloniki). Did you try (if applicable) to average spatially the values from the stations within one grid cell and see if it reduces some of the discrepancy in the representativeness between model and point measurements?

Two pairs of air quality stations in Thessaloniki are indeed located in the same grid as can be now seen in the updated Figure 2, which includes the actual grid we are working with. We now include in our analysis the average observational levels of the two urban background stations (Malakopi and AUTH) that are situated in grid-pixel [22.95E, 40.625N] and the average of the urban industrial stations Sindos and Kordelio in grid–pixel [22.85E,40.975N] in Thessaloniki. However Figure 1 becomes very busy when the actual grid of the model run is plotted, and since the main purpose of Figure 1 is to depict the orography of the two areas and to give the reader a general idea of the regions of study and their surroundings, we opted to keep the original gridlines. We have included the following in the text:

"When more than one in-situ station is located at one grid-pixel their average value is considered, as a result in the case of Thessaloniki the mean values of the urban background stations "Malakopi" and "AUTH" is calculated as well in the case of the urban industrial stations "Kordelio" and "Sindos"."

[Figure]

4)        The discussion on the uncertainty is a bit qualitative sometimes. Many figures lack errorbars (see technical points below). For example, what are the uncertainties associated with individual max-doas measurements? If you use an average over time, could you include some estimation of the variability in this time range? Also, how much do you expect your geometric AMF calculation to change the result compared to the calculation that takes into account the actual NO2 profile? The same applies for the comparison with TROPOMI NO2. A more quantitative description of the uncertainties would also help in understanding how the discrepancies you find compare with these uncertainties

Following the reviewer's suggestions, error bars referring to the standard deviation of the averaged MAX-DOAS observations and LOTOS-EUROS simulations are added to Figures 7 and 10. Further, new figures were added to the updated text which show the diurnal variability of the in-situ measurements and the LOTOS-EUROS simulations including their standard deviation as a shaded area (see Figures S3, S9 and S10 in the supplement).
The following comment on the uncertainties associated with the geometric AMF calculation is added in the manuscript:
"The evaluation of the magnitude of the differences introduced by using the geometric AMF instead of a full AMF calculation is ongoing for both these instruments. We mention here the work of Shaiganfar et al., 2011, who reported that tropospheric NO2 columns deviate by approximately ±20% for NO2 layer heights≤500m and a moderate aerosol optical depth, when using the geometric approximation instead of a full AMF calculation."

[Figure]

As far as the TROPOMI data are concerned, the tropospheric $NO_2$ precision field provided by the TROPOMI product is added at Figure S11, as a shaded area, to provide a more quantitative description of the variability of the TROPOMI observations.

[Figure]

5)      Sect. 3.2 Maybe I lost this information but which direction you use for AUTH: 1 or 2?

Initially one direction was used for the analysis but after the corrections in the manuscript the average of the direction 1 and 2 is used and the statistical analysis was similarly updated. A more clear comment about which direction is used is added to the manuscript:

"For this study, we used the average value of the two azimuth angles: 220° and 255° designated in Figure 2 by the purple lines 1 and 2, respectively."

6)      Sect. 3.3 Could you please clarify how do you apply the averaging kernels of TROPOMI to the model? How do you spatially and temporally collocate TROPOMI and the model? How do you interpolate vertically? Which level you use for the tropopause (from TROPOMI perhaps)?

The process of the implementation of averaging kernels onto LOTOS-EUROS model is made directly by a module of the model. It is true that it is not clear in the text how the averaging kernel are applied so a better description is added in the manuscript, as follows:

"The TROPOMI averaging kernels are applied onto the LOTOS-EUROS profiles using an online module of LOTOS-EUROS. After regridding the TROPOMI data onto LOTOS-EUROS gridding, the module maps the model profile to the retrieval a-priori layers, while in order to cover the atmosphere above the model's vertical levels boundary conditions are added from the CAMS NRT product. The averaging kernels are applied to the simulations made at the

[Figure]

closest time of the observations. The entire process is fully automated within the LOTOS-EUROS post-processing analysis tools."

Technical comments

7)	Table 7 you write in the caption: "The positive biases are shown in bold." But there is no bolded text in the table. Also, one horizonal line is missing.

The part in the caption about the positive biases is removed, as well as the line.

8)	Figure 7. The y-axis title of the second panel is not visible here

Thank you very much for noticing. The figure is changed.

9)	Figure 7 and 10 should have errorbars.

Error bars referring to the standard deviation of the averaged observations and simulations are added to figures 7 and 10 as recommended.

10)	Figure 9. Please write in the caption what the errorbars are.

Figure 9 now appears at the Supplement in Figure S6. Thank you very much for noticing that the error bars explanations is missing. These were added in Figure 8 and S6 and S7 as follows:

[Figure]

"The horizontal error bars refer to the standard deviation of averaged MAX-DOAS observations while the vertical error bars refer to the standard deviation of averaged LOTOS-EUROS simulations"

11)      Overall, the paper is a bit figure and table -heavy in my opinion and a bit repetitive sometimes. Maybe you can try to shorten some text throughout. For example, while it is useful to have these summaries at the end of each paragraph, it could be written in a more concise manner. Also, some of the tables and figures could go to the supplement. For example, Fig. 4, Fig. 13, Table 7 could be moved to the supplement. Perhaps some figures could be also grouped together. Following the reviewer's suggestions new sections were added in the section 3, where results are shown, to make the discussion more concise. The added sections are:

 3.1.1     Discussion on the validation of surface $NO_2$ concentrations

 3.2.3     Discussion on the validation of tropospheric $NO_2$ columns against ground-based MAX-DOAS observations

3.3.1      Discussion on the validation of tropospheric $NO_2$ columns against S5P/TROPOMI observations

Finally some Figures and Tables are moved to the supplement (i.e. . Figure 9 and 13) or completely removed (i.e. Table 3, Figure 9) in order to make the paper less busy.

---

## Author Comment (AC2) · 8 Feb 2021

The paper "Evaluation of the LOTOS-EUROS NO2 simulations using ground-based measurements and S5P/TROPOMI observations over Greece" by Skoulidou et al., presents LOTUS-EUROS NO2 simulation over Greece, for a period of 7 months, from June to December 2018. The data are compared to in-situ NO2 concentrations for the whole period at 14 sites in Athens and Thessaloniki, then to 2 months (June and December) of tropospheric NO2 VCD from MAX-DOAS instruments in Athens and Thessaloniki and from TROPOMI. Differences as a function of the season are discussed. The scientific content of the paper fits the scope of ACP, and the paper is interesting,

although a bit difficult to read due to length and repetition in several sections. The number of tables and figures could also be reduced. It is a pity that profiles are not exploited a bit more. It would be nice to have: 1) comparisons of the LOTOS-EUROS NO2 profiles wrt to TROPOMI a-priori profiles from TM5, and 2) at the MAX-DOAS stations, profiles retrievals (instead of only tropospheric VCD from geometrical approximation) could be used as a link between the surface NO2 from the in-situ and the tropospheric VCD from TROPOMI. Moreover, the LOTUS-EUROS profiles could be used to test a relation between surface and tropospheric VCD, and test this assumption with the insitu and MAX-DOAS measurements. I would thus recommend some revision of the text and some further investigations, as described below. specific comments:

Following the reviewer's suggestions new sections were added in the section 3, where results are shown, to make the discussion more concise. The added sections are:

3.1.1     Discussion on the validation of surface $NO_2$ concentrations

3.2.3     Discussion on the validation of tropospheric $NO_2$ columns against ground-based MAX-DOAS observations

3.3.1     Discussion on the validation of tropospheric $NO_2$ columns against S5P/TROPOMI observations

Moreover the conclusions are changed to Conclusion and Discussion and rewritten in order

to make the take home message clearer.

Finally some Figures and Tables are moved to the supplement (i.e. Figure 9 and 13) or completely removed (i.e. Table 3, Figure 9) in order to make the paper less busy.

The specific comments are answered below:

- why not compare also to MAX-DOAS from June "to" December (as for the in-situ) instead of June "and" December?

The aim of comparing the LOTOS-EUROS surface concentrations against n-situ surface concentrations over a period 7 months is to estimate the ability of LOTSO-EUROS to simulate sufficiently well the near-surface concentrations, where the main $NO_2$ sources are located, over the Greek domain. Furthermore, we wanted to study the performance of the simulated VCD of the model over a typical summer month (i.e. July) and a typical winter (i.e. December) month using MAX-DOAS and TROPOMI observations. Indeed, we understand that this might cause confusion to the reader so in order homogeneity in all comparisons shown, we report comparisons to the in-situ measurements only for a summer and a winter month.

- Are MAX-DOAS profiles available? they could make the link between surface NO2 values from in-situ and tropospheric VCD

The MAX-DOAS profiles over Thessaloniki are not available for this study and the tropospheric

[Figure]

VCD is calculated with a geometric approximation. To ensure a homogeneous analysis and to compare similar kind of MAX-DOAS data to the LOTOS-EUROS simulations, we used the geometric approximation VCDs for Athens as well. Comparisons between MAX-DOAS products and in-situ observations are currently in progress by the MAX-DOAS team of our laboratory and form the focus of a separate, MAX-DOAS-focused publication.

-        Also how are the LOTOS-EUROS NO2 profiles comparing to TROPOMI a-priori profiles from TM5?

As stated at the manuscript the TROPOMI data become independent of the a-priori profile shapes of TM5-MP model when the averaging kernels are used. In our study the averaging kernels of TROPOMI are applied in the model profiles and as a result the comparison of the satellite data with LOTOS-EUROS simulations is not influenced by the retrieval a-priori, as clearly stated in page 22 of the PUM (https://sentinels.copernicus.eu/documents/247904/2474726/Sentinel-5P-Level-2-Product-User-Manual-Nitrogen-Dioxide). For this reason we consider that the comparison of LOTOS-EUROS profiles with TM5 profiles will be not so relevant in this study but maybe will be very interesting for a different study where averaging kernels are not applied, or are calculated in a different manner (also suggested in page 22 of the PUM).

-        how are the TROPOMI AVK applied to the LOTUS-EUROS model? It is said that gridded data are created from TROPOMI pixels. Are the AVK averaged to created a grid of AVK?

The process of the implementation of averaging kernels onto LOTOS-EUROS model is made directly by a module of the model. It is true that it is not clear in the text how the averaging

kernel are applied so a better description is added in the manuscript, as follows:

"The TROPOMI averaging kernels are applied onto the LOTOS-EUROS profiles using an online module of LOTOS-EUROS. After regridding the TROPOMI data onto LOTOS-EUROS gridding, the module maps the model profile to the retrieval a-priori layers, while in order to cover the atmosphere above the model's vertical levels boundary conditions are added from the CAMS NRT product. The averaging kernels are applied to the simulations made at the closest time of the observations. The entire process is fully automated within the LOTOS-EUROS post-processing analysis tools."

P4, L 114: there are 10 levels "from the surface to a top around 175 hPa (about 12 km)." Are all the levels of same width?
It is more clearly added at the manuscript that the levels are coarsening upwards.

[Figure]

P4, L 150: "For this reason, stations characterized as urban traffic stations, localised close to busy traffic roads of the city and showing very large values, are excluded from the validation." –> how this selection is done? we still have urban traffic sites in Sect 3.1...

The official designation of the station type was assumed to be that one reported in the official databases, however due to our detailed knowledge of where those stations are located we decided to exclude ones that are exactly over busy thoroughfares in Athens. It follows that those stations are directly affected by the smallest changes in road emissions and their reported measurements far noisier than stations that are within the city canopy but not directly on a busy road. We have included the following sentence in the text:

"For this reason, stations characterized as urban traffic stations, localised close to busy traffic roads of the city and showing extremely high concentrations were excluded from the validation, based on local knowledge of their actual locations. As a result, we include in our analysis stations that are officially characterized as "traffic stations" (e.g. Marousi station, Athens) but which are not placed directly over the major thoroughfares."

P5, L.160: why only "for July and December" and not between July and December as for the in-situ comparisons (or over the whole year)?
Many of the stations that we used for the in-situ measurements had large gaps for the period between January and May and as a result we excluded these months for the total of the 14 stations. Further, we wanted to estimate the ability of LOTOS-EUROS to simulate VCDs for a typical summer and winter months, in our case July and December, and for this

reason we made our analysis for these months. The main scope of this paper was the validation of the surface simulations with in-situ measurements. However, after the reviewer's comments, we found it necessary to change our analysis and consider for the in-situ measurements July and December as the summer and winter period and not July-August or November-December as we first did, to succeed a better consistency between the surface and VCD analysis.

[Figure]

P5, L.165: why the 15° elevation has been chosen and not the 30° elevation? is there any further selection, as in Drosoglou et al., 2017 where an average of both elevations was considered if the results from the 2 angles are within 20 or 30% ?

In our case we used the 15° degrees elevation angle because the results are considered by the MAX-DOAS teams/co-authors to be more representative of the city as they probe air closer to the instrument. Also, the quality of the data is improve for the case of this elevation angle due to an improved single-to-noise ratio. Note that the same elevation angle is used at the official S5P validation (https://mpc-vdaf-server.tropomi.eu/no2/no2-offl-maxdoas/athens#Comparison) for the MAX-DOAS instrument in Athens.

P5, L.166: both the azimuth 220° and 255° are looking over the sea. Do you have viewing directions in the same direction than the in-situ measurements (from AUTH to Lagada and Kordelio (this last name and Sindos are difficult to read in Figure 2))?

Unfortunately, no different azimuthal angles are available from the MAX-DOAS in Thessaloniki, at least for our period of interest. We changed the plot hoping that it is easier to read now.

P5, L.181: for Athens, the geometrical approximation is also used, but from which elevation angle?

also 15°, as for Thessaloniki?

Exactly, for Athens as well the 15 degrees are chosen for the same reasons that are stated

before. It is true it is not clear in the text we tried to make it clearer in the manuscript adding:

"For this study, MAX-DOAS observations at 15° elevation angle are analysed in both sites in order to avoid uncertainties introduced due to aerosols at lower elevation angles (Sinreich et al., 2005), to obtain a stronger signal-to-ratio and since this elevation angle probes air closer to location of the emissions, to ensure a stronger signal overall. "

P5: please specify if MAX-DOAS data are filtered for clouds, and give an estimation of the errors on the tropospheric VCD for both sites. Are they of equivalent quality? Please also discuss the MAX-DOAS horizontal representativeness area (or at least mention the outcome from Drosoglou et al., 2017 and Gratsea et al., 2016). Are these taken into account in the comparison, or is the MAX-DOAS considered as a "point measurement" in the horizontal plane?

The observations for Thessaloniki are not filtered for clouds. In Athens they are not explicitly filtered for clouds either, however there is a criterion for agreement between the measurements in their two operational observing elevation angles (15° and 8°). This criterion imposes an implicit filter for viewing conditions where the geometric approximation is appropriate which basically excludes days with broken clouds.

The instruments are not of equivalent quality mostly due to the different signal-to-noise ratio of the spectrometers included in each instrument suite, with the Athens MAX-DOAS

having an improved performance.

Comment on the uncertainties added by the retrieval algorithm is added in the manuscript:

"The evaluation of the magnitude of the differences introduced by using the geometric AMF instead of a full AMF calculation is ongoing for both these instruments. We mention here the work of Shaiganfar et al., 2011, who reported that tropospheric $NO_2$ columns deviate by approximately ±20% for $NO_2$ layer heights≤500m and a moderate aerosol optical depth, when using the geometric approximation instead of a full AMF calculation."

The MAX-DOAS horizontal representativeness area and the selection of the simulations grid-cells that are chosen for the comparisons are now discussed in the paper:

For Thessaloniki:

" Drosoglou et al., (2017) found that the MAX-DOAS instrument in AUTH has an average representative distance of 0.55 km which can be as high as 10 km during spring and reach even longer distances in summer. During a campaign period, that took place in late autumn to spring, when the height of boundary layer is low, Drosoglou et al. (2017), found that only 2% of the data exceed the 2 km horizontal distance. Consecutively, in our analysis the simulation grid-cell where the MAX-DOAS is situated is considered as the most appropriate to compare with the observations."

For Athens:

"Two azimuthal viewing angles are selected in this case as well, at 120° and at 232.5°, and

are represented by the purple lines Figure 2. The first one, marked with "R" is characterized as a rural unobstructed direction, while the other one is named "U" and views towards an urban direction and as a result the simulations of the closest grid-cells that are in the "U" and "R" directions and are representative of urban and rural areas respectively are selected for the comparisons. The horizontal representativeness of the instrument is estimated to be ~2 km while comparisons with in-situ $NO_2$ measurements have shown that areas close to the instrument are better represented though this premise has not yet been quantified."

P6, Fig2: in this figure, several in-situ stations are in the line of sight of one of the MAX-DOAS azimuth direction. Do they show similar diurnal variation? or can these be compared using the model profile shape to convert surface to tropospheric VCD? Similarly, adding the extension of the LOTOS-EUROS $0.1°×0.05°$ grid on these maps, could help the reader understanding if several in-situ stations are in one model grid cell.

Following the reviewer's suggestions we plotted Figure 2 using the actual grid $0.1°×0.05°$. The diurnal variations of the in-situ stations that are in the same direction with MAX-DOAS in Thessaloniki and Athens are plotted for both July and December between 6 and 13 UTC, same as for the MAX-DOAS plots, and can be seen in Figures S9 and S10.

For the region of Athens and the urban direction station "Geoponiki" is shown in Figure S9: "This assumption was further confirmed where the diurnal variation between the surface LOTOS-EUROS $NO_2$ simulations and the in-situ measurements of the suburban industrial

station "Geoponiki", located at the same pixel of the Athens MAX-DOAS urban direction, is examined (Figure S9). The surface simulations overestimate the NO$_2$ concentrations at 6 UTC while for the remaining hours the model underestimates the in situ measurements. "

For the region of Thessaloniki the average stations AUTH/Malakopi are shown in Figure S10. The following text has been added to the updated paper:
" Figure S10 shows the diurnal variations of the surface LOTOS-EUROS NO$_2$ simulations at the same grid-cell, as the MAX-DOAS, for July [upper] and December [lower] together with the surface measurements of the in-situ stations in the area. They show a similar variation with the MAX-DOAS and simulated columns, overall underestimating all hourly in situ measurements. However, during the early hours in July the model simulates higher NO$_2$ pointing to an overestimation of the surface simulations that is not present in the columnar comparisons and may be attributed to a low boundary layer assumed in the simulations."

[Figure]

P6, L..203: "that because the averaging kernels are used in our case, the comparison with LOTOS-EUROS is not influenced by the retrieval a-priori (Eskes and Boersma, 2003" –> this is the case for LOTOS-EUROS vs TROPOMI, but not eg for the LOTOSEUROS vs MAX-DOAS comparisons. Moreover, this is a bit misleading, as the coarse apriori model profiles would still play a role. Are the TROPOMI AVK also gridded "onto the LOTOS-EUROS grid at 0.1°×0.05°." (P.6, L 208)? How are the AVK applyed?

Please explain.

Averaging kernels are not applied to the geometric approximation which has been used for the derivation of the VCDs. The assumption is of course that all altitudes contribute similarly. In the case of the comparisons between the TROPOMI data and the LOTOS-EUROS simulations the TROPOMI averaging kernel are applied onto the model profiles making the comparison independent of the a-priori profiles, as stated by Eskes and Boersma (2003). A more clear comment about the application of the averaging kernels was added in the manuscript:

"The TROPOMI averaging kernels are applied onto the LOTOS-EUROS profiles using an online module of LOTOS-EUROS. After regridding the TROPOMI data onto LOTOS-EUROS gridding, the module maps the model profile to the retrieval a-priori layers, while in order to cover the atmosphere above the model's vertical levels boundary conditions are added from the CAMS NRT product. The averaging kernels are applied to the simulations made at the closest time of the observations. The entire process is fully automated within the LOTOS-EUROS post-processing analysis tools."

[Figure]

P7, Sect 3.1: what is the width of the first model level, that is compared to the in-situ surface concentrations? The discussion would be more easy to follow if instead of the tables 1 and 2 (or in addition, maybe in the supplement) a few plots of the diurnal variation of the in-situ and the model at the sites is shown (a bit like figures 7 and 10). It would be more easy to also understand why the 12-15pm is selected as representative of "daytime" conditions and 0-3am of "night" condition. Are there big changes outside these periods? It would also allow to draw a conclusion on the consistency (or not) of the diurnal surface NO2 variation compared to the diurnal NO2 VCD variation.

For the vertical structure, the model uses the level layers of the 137 hybrid sigma-pressure layers used by ECMWF for the operational meteorological forecasts. The width of the first layer is around 25 meters.

Representative plots of the diurnal variation at 3 air quality stations and the LOTOS-EUROS simulations are added in the supplement of the manuscript in Figure S3 and shows the distinct behavior of the model during nighttime and the persistent underestimation during daytime period. As commented in the paper:

"The diurnal $NO_2$ variability of both the LOTOS-EUROS simulations (in red) and the corresponding measurements for three air quality stations (in black) are presented in Figure S3, where it is shown that the model simulates very well the expected highs and lows of the $NO_2$ concentrations during the day, with some differences in the absolute levels, whose

possible origins are discussed in Section 3.1.1"

Figure 7 and 10: what would be the MA X-DOAS if retrieved from 30° elevation instead of 15°?

For this study, we used MAX-DOAS observations at 15° elevation angle in both sites in order to avoid uncertainties introduced due to aerosols at lower elevation angles (Sinreich et al., 2005), to obtain a bigger signal and as a result a better quality of data and to be more representative of the city values as they probe air closer to the instrument.

P20, Sect4: the main messages are a bit lost in the conclusion, which is a bit too much a repetition of each subsection conclusions.

The conclusions are rewritten accordingly to make the take home message clearer.

Technical comments and corrections:

————————————————-

P2, L.71: (Zerefos et al., 2000.) –> (Zerefos et al., 2000).

Thank you for noticing, we have corrected the mistake.

P4, L.117: what is the "tree-species database"?

Added to the manuscript the above clarification:

[Figure]

"Biogenic emissions (isoprene) are calculated online using the meteorology and a detailed land use and tree-species database that contains 115 tree species and their biomass density and emission factors for terpene and isoprene that allows the emission calculation per tree species type."

P4, L.135: 2.2Ground-based –> add a space after the section number

Thank you for noticing, I added a space

P4, L.143: give some references for the chemiluminescence method
A reference and a short explanation are added:

"A chemiluminescence measurement method, which is the method based on the reaction of ozone with nitric oxide to form excited NO2 that emits infrared light (Dunlea et al., 2007 ), is used for the measurement of nitrogen dioxide concentrations at the stations"

[Figure]

P5, L.172: quantify "good agreement"

The mean biases calculated by Drosoglou et al are included to quantify the agreement:

"In the study of Drosoglou et al. (2017), the tropospheric $NO_2$ derived from MAX-DOAS instruments positioned at three different locations around Thessaloniki and the OMI/Aura satellite were compared during a 6-month campaign showing good agreement over the rural and the suburban areas with a mean bias of -1.63 $molec.cm^{-2}$ and -0.17 $molec.cm^{-2}$ respectively."

P6, L.194: "several TROPOMI NO2 validation papers that have been recently submitted (Judd et al., 2020; Verhoelst et al., 2020)." –> there are some TROPOMI NO2 validation papers already fully published: Zhao et al 2019 (https://www.atmos-meas-techdiscuss.780net/amt-2019-416/ ), Ialongo et al., 2020 ( https://doi.org/10.5194/amt-13-205-2020 )

Thanks for the comment. We referred to the papers you suggest. Moreover we made a reference to the results of Zhao et al., 2020,  that weren't introduced:

"This same feature was also shown by Zhao et al., (2020) who compared the TROPOMI total columns with Pandora total columns in Greater Toronto Area over an urban and suburban site and found negative biases while the same comparisons over a rural site showed positive bias"

[Figure]

P7, L.229: "the correlations are very good (r=0.69 and 0.63" –> remove "very"
The word "very" is removed and the phrase is changed to:
"AUTH/Malakopi" the correlation is found to be best (r=0.68 and r=0.72) during daytime and winter respectively

P7, L.234: "In this case, a clear seasonal pattern in the model's performance, as is the case for Thessaloniki, was not found." –> do you have any hint why?

After following reviewer's comment we changed our analysis in order to ensure a better homogeneity in our comparisons of in-situ measurements with VCDs. For this reason the summer in the in-situ comparisons now refers only to July month (initially July and August) and winter only to December (initially November and December). Our statistical results were similarly changed and the following comment has been added:

'As for Thessaloniki, in the case of Athens as well, the model shows improved correlations during the winter period than in summer with an average correlation of 0.53 and 0.46 respectively."

P7, Table 1 and 2: please add the units of the RMSE.

[Figure]

The RMSE is replaced by the NRMSE in order to be more informative for the comparison between different stations as recommended by Reviewer #3.

P10, L.283: please add in Figure S3 caption or ylabel, the definition of relative biases. (simulation-obs)/obs? Also, it could be nice to have a different symbol for each site, so that it would be clear for the reader which site(s) are the outliers of the whiskers in Winter and night conditions. Either 14 symbols, either grouped by station types introduced in tables 1 and 2, either one color per Athens, one per Thessaloniki...

Thank you for noticing that a label was missing. We added one at the caption of figure which is placed now in the main manuscript in Figure 5. Moreover we follow the reviewer's comments and we annotated the stations in different according to their type:  red in the case of urban stations, blue in the case of traffic and green for suburban.

P10, L.394: please specify how "spatial correlation coefficient" and "temporal correlation" are calculated.

We specified the correlation coefficients we used in the manuscript as bellow:

"Note that in this work temporal correlation refers to the correlation between the average daily values of TROPOMI and the simulations in each region while spatial correlation refers to the correlation between the monthly average observations and the corresponding simulations in each grid cell of the region."

[Figure]

P10, L.298: "representation issues related to the location of the stations" –> link with the Drosoglou et al., 2017 study with high resolution model (6km resolution for the Balkans and 2km resolution for the Thessaloniki region)

According to the reviewer's comment a link between our results and Drosoglou et al.,2017, is added in the manuscript. It is considered more appropriate to add the comment at the part when we apply the comparison with the MAX-DOAS measurements:

"The 0.1x0.05° pixel covers a relatively large area of the city of Thessaloniki and inadvertently includes some inhomogeneous air pollution patterns, since it engulfs both the city centre as well as the surroundings, cleaner, areas, while the MAX-DOAS probes air straight from the city centre (Drosoglou et al, 2017). As a result, the relatively large grid pixel of the model simulation might underestimates a possible horizontal plume from industrial areas, such as that from chimneys."

P11, L/ 320: "The MAX-DOAS in the center of Thessaloniki observes high NO2 columns during the winter months and lower levels during the spring season..." –> is this a description of the rest of the MAX-DOAS dataset, not shown here, or is "spring season" mis-referring to July data or to the Drosoglou 2017 results? If other months than December and July are available from the MAX-DOAS, how are they comparing to LOTUS-EUROS?

Spring season is referring to the results of Drosoglou et al., 2017.

P11, L.237: "The daily mean" - how are the daily mean performed? is it , as for the

in-situ, only 12 to 15pm, or is it all the available points (below 75 ° SZA)? Is there a difference between the 2 approaches for MAX-DOAS data?

The daily mean in the case of the MAX-DOAS comparisons refers to the average value of the measurements and simulations in a day between the 6:00 and 13:00 UTC time. When gaps appear in the measurements are considered in the simulations as well and are not taken into account in the analysis.

[Figure]

P12, L.355: are the MAX-DOAS data cloud filtered? if there are some gaps in the MAX-DOAS, are these gaps considered also in the model data, before doing the daily average?

The MAX-DOAS data are not filtered for clouds. The MAX-DOAS data have been averaged to hourly data in order to be comparable with the LOTOS-EUROS outputs. However the comparisons that are applied in the analysis consider LOTOS-EUROS simulations only at the hours when MAX-DOAS measurements are available. If any gaps appear throughout the measurements it follows that the equivalent LOTOS-EUROS data are not considered as well.

Figure 7 and 10: please increase a bit the size of these figures. The legend is diffcult to read.

Thank you for noticing. Figure 7, 10 and S6 have been replaced using larger fontsizes in the legends

P15, L.423: "the MAX-DOAS tropospheric columns in both cities have been derived using the geometric approximation without taking into account the actual NO2 profile, introducing therefore, additional uncertainty" –> please estimate this error.

The main uncertainty from using the geometric approximation instead of a full AMF treatment including aerosols etc. has been addressed by certain studies looking into this issue, for e.g. Shaiganfar et al. (2011). According to this we added the following reference in our text:

"The evaluation of the magnitude of the differences introduced by using the geometric AMF instead of a full AMF calculation is ongoing for both these instruments. We mention here

the work of Shaiganfar et al., 2011, who reported that tropospheric $NO_2$ columns deviate by approximately ±20% for NO2 layer heights≤500m and a moderate aerosol optical depth, when using the geometric approximation instead of a full AMF calculation."

P15, L424: "the one azimuthal directional observation in Athens compared with a grid cell of the model may not be representative of the relatively large grid pixel of the model simulation, underestimating a possible horizontal plume from industrial areas i.e. from chimneys" –> mention and discuss a bit more the MAX-DOAS horizontal representativity and the model size.

The MAX-DOAS horizontal representativity has been discussed following the reviewer's recommendations.

For Thessaloniki:

" Drosoglou et al., (2017) found that the MAX-DOAS instrument in AUTH has an average representative distance of 0.55 km which can be as high as 10 km during spring and reach even longer distances in summer. During a campaign period, that took place in late autumn to spring, when the height of boundary layer is low, Drosoglou et al. (2017), found that only 2% of the data exceed the 2 km horizontal distance. Consecutively, in our analysis the simulation grid-cell where the MAX-DOAS is situated is considered as the most appropriate to compare with the observations."

[Figure]

For Athens:

"Two azimuthal viewing angles are selected in this case as well, at 120° and at 232.5°, and are represented by the purple lines in Figure 2. The first one, marked with "R" is characterized as a rural unobstructed direction, while the other one is named "U" and views towards an urban direction and as a result the simulations of the closest grid-cells that are in the "U" and "R" directions and are representative of urban and rural areas respectively are selected for the comparisons. The horizontal representativeness of the instrument is estimated to be ~2 km while comparisons with in-situ $NO_2$ measurements have shown that areas close to the instrument are better represented though this premise has not yet been quantified."

P15, L.434: "The averaging kernels are applied directly by the LOTOS-EUROS model"
–> "by" or "to" ? Explain better how the AVK are applied (gridded AVK? application of AVKK at the pixel level, and then gridding? ...?)

The process of the implementation of averaging kernels onto LOTOS-EUROS model is made directly by a module of the model. It is true that it is not clear in the text how the averaging kernel are applied so a better description is added in the manuscript, as follows:

"The TROPOMI averaging kernels are applied onto the LOTOS-EUROS profiles using an online module of LOTOS-EUROS. After regridding the TROPOMI data onto LOTOS-EUROS gridding, the module maps the model profile to the retrieval a-priori layers, while in order to

[Figure]

cover the atmosphere above the model's vertical levels boundary conditions are added from the CAMS NRT product. The averaging kernels are applied to the simulations made at the closest time of the observations. The entire process is fully automated within the LOTOS-EUROS post-processing analysis tools."

P15, L.447: Pandora measurements in Helsinki are total columns!

Thank you very much for noticing we added in the text that we refer to total columns so to not lead to any misleading.

P15, L 452: "there is no NO retrieval profile-related bias influencing the comparisons" –> NO to NO2 this is partially true, but the influence of the coarser TM5 1x1 degree resolution instead of a regional high resolution model is still present (see Zhao et al., 2019).

Zhao et al. (2020) compared the standard TROPOMI product and a TROPOMI new product in which they used a high resolution regional air quality forecast model in the AMF calculation against Pandora measurements in three different sites. They conclude that their new product is an improvement over the standard TROPOMI products. However, from our understanding they did not apply the TROPOMI averaging kernels during their analysis. In our case the TROPOMI data become independent of the a-priori profile shapes of TM5-MP model because we apply the averaging kernels at the model profile (Boersma and Eskes, 2003) , as clearly recommended in page 22 of the PUM (https://sentinels.copernicus.eu/documents/247904/2474726/Sentinel-5P-Level-2-Product-

[Figure]

User-Manual-Nitrogen-Dioxide).

[Figure]

P15, L.455: "the profiles of LOTOS-EUROS peak more strongly near the surface" –> it would be interesting to see the comparison of the profiles shapes (TM5 vs LOTOSEUROS).
In our case the TROPOMI data become independent of the a-priori profile shapes of TM5-MP model because we apply the averaging kernels at the model profile, as discussed previously. A comparison of the profile shapes between the two models is rather beyond the scope of this paper.

P17, L.488: "In December (Figure 12, middle panels) LOTOS-EUROS simulates high NO2 columns (mean value ~5×1015 molec.cm2) near the Isthmus of Corinth, which are not supported by the TROPOMI observations, pointing to a possible overestimation of the NOx emissions in the area" - it could maybe also be related to winds that do not add up? It would be nice to see the TROPOMI December map if the winds speed and direction would be taken into account to create the map (Zhao et al., 2019; Lorente et al., 2019)

It would be indeed very interesting to perform such an analysis. We are currently working on our new publication where we estimate new NOx emissions using LOTOS-EUROS CTM to assimilate TROPOMI data over regions with high anthropogenic emissions (mostly where power plants are located).We further aim to apply a wind-rotation to explain our findings. To avoid confusion with this work, we have opted to exclude this sentence from our updated manuscript.

[Figure]

Lorente et al., 2019: https://www.nature.com/articles/s41598-019-56428-5

Zhao et al 2019: https://www.atmos-meas-tech-discuss.780net/amt-2019-416/

Ialongo et al., 2020: https://doi.org/10.5194/amt-13-205-2020

―――――――――――――――――――――――

---

## Author Comment (AC3) · 8 Feb 2021

The authors compare the LOTOS-EUROS simulations of NO2 over Greece against surface measurements, DOAS profiles and sentinel maps during the second half of 2018. The comparison is also performed at different seasons, sites and hour of the day, and the authors provided reasoning for the differences. The paper is within the scope of the journal and it is scientifically sound. My main concern is on the significance of some results, which affects its emphasis and extent. I trust it should be published, following the recommendations hereafter.

Specific Comments 1. The validation approach relies mainly on correlation and RMSE.

The linear correlations should be tested for their significance. The same applies also for the spatial correlations, for which, the estimation algorithm is missing. Use of NRMSE is more informative when comparing the simulations at different stations.

The sum of the linear correlations are tested as commented by the reviewer for their significance calculating the p-value and the results found are commented throughout the manuscript. In the case of the comparisons of the simulations with in-situ data we found that the p-value is in all cases much lower than the significance level (a=0.05) and the correlation is statistically significant as added in the updated manuscript. In the case of the MAX-DOAS comparison with LOTOS-EUROS simulations in the region of Thessaloniki in July and December the correlation coefficients are statistically significant for a=0.05. In Athens it was found that only over the rural direction in July the correlation coefficient is not statistically significant. When comparing TROPOMI and LOTOS-EUROS, the spatial correlations are found statistically significant for all the regions and periods while the temporal correlations over Thessaloniki in July and Greece in December found statistically not significant.

Moreover the estimation algorithm of the spatial and temporal correlation is added in the text as recommended by the reviewer:

"Note that in this work temporal correlation refers to the correlation between the average daily values of TROPOMI and the simulations in each region while spatial correlation refers to the correlation between the monthly average observations and the corresponding simulations in each grid cell of the region."

Further the NRMSE instead or RMSE is used at tables 1 and 2 following reviewer's recommendation.

2. Can the authors comment on the impact of the 24h periodicity to the temporal correlations?

Diurnal variation plots of representative stations of the in-situ stations are added in the supplement (S3, S9 S10)

3. The way and reason some stations have been excluded should be re-framed to be less qualitative.

The official designation of the station type was assumed to be that one reported in the official databases, however due to our detailed knowledge of where those stations are located we decided to exclude ones that are exactly over busy thoroughfares in Athens. It follows that those stations are directly affected by the smallest changes in road emissions and their reported measurements far noisier than stations that are within the city canopy but not directly on a busy road. We have included the following sentence in the text:

"For this reason, stations characterized as urban traffic stations, localised close to busy traffic roads of the city and showing extremely high concentrations were excluded from the validation, based on local knowledge of their actual locations. As a result, we include in our analysis stations that are officially characterized as "traffic stations" (e.g. Marousi station, Athens) but which are not placed directly over the major thoroughfares."

4. The comparison of the gridded LOTOS-EUROS simulations against point measurements needs some clarifications. Ideally, one should either compare the observations with the simulations pin-pointed at the station location or the model grid values with the cluster of observations falling inside.

Two pairs of air quality stations in Thessaloniki are indeed located in the same grid as can be now seen in the updated Figure 2, which includes the actual grid we are working with. We now include in our analysis the average observational levels of the two urban background stations (Malakopi and AUTH) that are situated in grid-pixel [22.95E, 40.625N] and the average of the urban industrial stations Sindos and Kordelio in grid–pixel [22.85E,40.975N] in Thessaloniki. However Figure 1 becomes very busy when the actual grid of the model run

is plotted, and since the main purpose of Figure 1 is to depict the orography of the two areas and to give the reader a general idea of the regions of study and their surroundings, we opted to keep the original gridlines.

5. Uncertainty estimates require a more rigorous framework, with a better description

Following the reviewer's comments on the uncertainty estimates we used error bars in Figures 7 and 10 referring to the standard deviation of the averaged MAX-DOAS observations and LOTOS-EUROS simulations. Furthermore, new figures added, according to another reviewer's comments, showing the diurnal variability of the in-situ measurements and simulations with their standard deviation of the averaged values as a shaded area (Figures S3, S9 and S10). Moreover, the tropospheric $NO_2$ precision of the TROPOMI data provided by the TROPOMI product has now been added at Figure 13 in the shaded area to provide a more quantitative description of the TROPOMI observations.

6. The comparison of the gridded LOTOS-EUROS simulations against satellite data needs some clarifications on the TROPOMI data regridding and the application of the averaging kernel in LOTOSEUROS.

The process of the implementation of averaging kernels onto LOTOS-EUROS model is made directly by a module of the model. It is true that it is not clear in the text how the averaging kernel are applied so a better description is added in the manuscript, as follows:

"The TROPOMI averaging kernels are applied onto the LOTOS-EUROS profiles using an online module of LOTOS-EUROS. After regridding the TROPOMI data onto LOTOS-EUROS gridding, the module maps the model profile to the retrieval a-priori layers, while in order to cover the atmosphere above the model's vertical levels boundary conditions are added from the CAMS NRT product. The averaging kernels are applied to the simulations made at the closest time of the observations. The entire process is fully automated within the LOTOS-EUROS post-processing analysis tools."

Technical Comments

[Figure]

Tables: Please specify which correlations are significant.

Following the reviewer's comments we have added which correlations are significant throughout the text.

 Figures: The information in some figures is not easily seen (e.g. Figure 4, 5).

Some figures are changed in order to be more easy to read them. As an example, Figure 4 is made larger (Figure 3 in the new changed manuscript), in Figure 5 (Figure 6 in new manuscript).